# Dynamic chromatin architecture of the porcine adipose tissues with weight gain and loss

Long Jin [1,2,13], Danyang Wang[3,4,5,13], Jiaman Zhang [1,13], Pengliang Liu[1], Yujie Wang [1], Yu Lin[1], Can Liu[1], Ziyin Han[1,6], Keren Long[1,2], Diyan Li [7], Yu Jiang [8], Guisen Li [9], Yu Zhang[1], Jingyi Bai[1], Xiaokai Li[1], Jing Li [1,2], Lu Lu[1,2], Fanli Kong[1], Xun Wang[1], Hua Li[6], Zhiqing Huang [10], Jideng Ma[1,2], Xiaolan Fan[1,2], Linyuan Shen[1,2], Li Zhu[1,2], Yanzhi Jiang [1], Guoqing Tang[1,2], Bin Feng [10], Bo Zeng[1,11], Liangpeng Ge[12], Xuewei Li[1,2], Qianzi Tang [1,2], Zhihua Zhang [3,4] ✉ & Mingzhou Li [1,2] ✉

Using an adult female miniature pig model with diet-induced weight gain/weight loss, we investigated the regulatory mechanisms of three-dimensional (3D) genome architecture in adipose tissues (ATs) associated with obesity. We generated 249 high-resolution in situ Hi-C chromatin contact maps of subcutaneous AT and three visceral ATs, analyzing transcriptomic and chromatin architectural changes under different nutritional treatments. We find that chromatin architecture remodeling underpins transcriptomic divergence in ATs, potentially linked to metabolic risks in obesity development. Analysis of chromatin architecture among subcutaneous ATs of different mammals suggests the presence of transcriptional regulatory divergence that could explain phenotypic, physiological, and functional differences in ATs. Regulatory element conservation analysis in pigs and humans reveals similarities in the regulatory circuitry of genes responsible for the obesity phenotype and identified non-conserved elements in species-specific gene sets that underpin AT specialization. This work provides a data-rich tool for discovering obesity-related regulatory elements in humans and pigs.

The global obesity epidemic poses a major threat to human quality of life and modern healthcare systems worldwide[1,2]. As much as 58% of the world's adult population is predicted to be overweight or obese by 2030[2]. The metabolic risk factors for obesity are more closely related to adipose distribution than to total adipose mass[3], which could be due to substantial differences in the contributions of anatomically distinct adipose tissues (ATs) to energy balance and nutrient homeostasis, as well as differences in the mechanisms by which distinct AT populations expand during obesity development. Individuals with obesity who display preferential expansion of visceral ATs (VATs) are at a greater risk for diabetes and cardiovascular disease than equally individuals

with obesity who store excess energy in subcutaneous ATs (SATs)[4]. In fact, the expansion of SATs can protect against metabolic complications related to high-energy feeding[3].

In recent decades, diverse animal models have been used to investigate obesity and its comorbidities, including small rodents, large animals (typically dogs and pigs), and non-human primates[5]. Due to profound similarities with humans in terms of their anatomical, physiological, and metabolic traits, domestic pig (*Sus scrofa*) models have enabled several major advances in metabolism research[5]. In particular, pigs have regional differences between anatomically distinct adipose depots (e.g., SATs and VATs), which are

---

highly similar to humans. This highlights the value of using pigs as obesity models[6].

Three-dimensional (3D) chromatin architecture is a fundamental regulator of transcription[7], and is organized in multi-scale hierarchical layers, including chromosome territories, compartments[8], topologically associating domains (TADs)[9], chromatin loops[10], and long-range interactions between promoters and enhancers (PEIs)[11]. An earlier study of chromatin architecture in adipose tissues provided early clues highlighting the regulatory importance of chromatin organization in adipogenesis[12]. Nonetheless, a panoramic view does not illustrate the dynamic changes in chromatin architecture that underpin transcriptomic divergence in ATs that are potentially linked to progressive metabolic risks in obesity development and dietary interventions.

To identify dynamic shifts in chromatin architecture related to obesity development, we used a miniature pig model of weight gain/ weight loss (WGWL) to generate a total of 249 high-resolution chromatin contact maps using in situ high-throughput chromatin conformation capture (Hi-C) sequencing and transcriptomes for four anatomically distinct ATs (one SAT and three VATs). In this WGWL model, the pigs were subjected to different nutritional conditions, including healthy pigs fed with a normal diet, obese pigs induced by a high-fat diet, and pigs subjected to dietary restrictions[13]. This relatively large-scale experiment enabled integrated analysis of the multi-scale reorganization of chromatin architecture and how it affects gene expression. Additionally, we comprehensively compared PEI organization in human and pig genomes across the four homologous ATs to characterize evolutionary divergence in their spatial regulatory circuitry and identify how that spatial rewiring could affect species-specific AT biology, such as the absence of brown AT in pigs. In addition to providing several insights into the functional divergence of ATs and the relative conservation of their 3D genomic regulatory mechanisms that support further exploration in pig models, this work also provides an important resource for future comparative metabolic research in humans and pigs.

## Results

### Transcriptome and chromatin architecture experiences alterations during WGWL in distinct ATs

We collected a total of 272 distinct AT samples from 68 adult female pigs across three nutritional condition groups. The normal condition (NC, $n = 12$) group served as the control group for healthy pigs fed a normal diet[14]. The weight-gain (WG, $n = 46$) group, fed with a high-fat diet for 22 weeks, exhibited a ~1.94-fold increase in body weight compared to the normal group, while the weight-loss (WL, $n = 10$) group (comprised of a subset of WG pigs subjected to a nutritional regimen of 10% of normal group caloric intake for 12 weeks) exhibited an average ~32.28% loss in body weight compared to the WG group at 22 weeks (Fig. 1a–d and see "Method" for details). The upper layer of backfat (ULB) at subcutaneous, and three abdominal VATs, i.e., greater omentum (GOM), mesenteric adipose (MAD), and retroperitoneal adipose (RAD), were collected (Fig. 1e). Analysis of ten representative metabolic indicators in serum samples revealed dysfunction in the metabolism[15,16] of WG pigs compared with NC animals, while these indicators were lower in the WL group than in the control group (i.e., the initial beneficial changes) (Supplementary Fig. 1a). Providing evidence for reports of higher plasticity in VAT size than in SAT size under variable nutrient intake[17,18], we observed a higher degree of hypertrophy in the adipocytes of three VATs between the NC and WG groups compared with the subcutaneous ULB adipocytes. In contrast, VATs in the WL group showed a greater degree of atrophy compared to ULB adipocytes (Fig. 1f and Supplementary Fig. 1b).

Examination of hierarchical 3D genome architecture with transcriptomic analysis revealed alterations in the WG and WL groups compared with the NC group. In situ Hi-C assays of all 249 AT samples generated a total of ~73.33 billion valid contacts (~294.52 million [M]

contacts per sample, reaching a maximum intra-chromosomal resolution of ~8 kb) (Supplementary Fig. 2, Supplementary Data 1 and Supplementary Note 1). Total RNA-seq for the corresponding AT samples indicated that the transcription, compartmental rearrangements, and local spatial context (reflected by insulation scores [IS]) diverged in the WG and WL groups compared to the NC group (Fig. 1g–l), suggesting that both excess and insufficient caloric intake can reshape chromatin architecture and transcriptomic patterns in ATs[19,20]. In particular, inflammation-related genes were differentially up-regulated in each AT type of the WG group, while metabolism-related genes were down-regulated in each WL group AT (Supplementary Note 1, Supplementary Fig. 3 and Supplementary Data 2). Notably, compartmentalization status showed that ten genes related to VAT hypertrophy[21] were in the more active compartment (i.e., more accessible) in WG compared to NC (Supplementary Fig. 4a), reflecting the enhanced absorption of free fatty acids and triglycerides in hypertrophic adipocytes after weight gain[17]. For example, increased expression of the tetraspanin family protein *TM4SF1* was previously reported in larger adipocytes of human subjects with obesity compared to its expression in smaller, non-obese subjects[21] (Supplementary Fig. 4b, c).

The t-SNE analysis found that transcriptomic profiles (Fig. 1g) and chromatin architecture, i.e., compartmentalization (Fig. 1h) and TAD organization (Fig. 1i) within ATs were more similar among groups, regardless of diet-induced alterations, than they were between ATs. This was especially true between ULB and VATs. Moreover, t-SNE plots comparing Hi-C and RNA-seq data for all groups showed distinct clustering of ATs, with a consistent order of similarity following ULB to RAD to MAD to GOM, while no such pattern appeared for the three nutritional conditions (Fig. 1g–i and Supplementary Figs. 5 and 6). This finding aligns with intrinsic functional and metabolic differences among ATs (Supplementary Fig. 7).

We next tested whether visceral RAD exhibits SAT-like metabolic characteristics[22] and found that RAD more actively regulates hyperplasia compared to visceral GOM or MAD. The compartmental status of *PPARG*, all nine *HOXD* cluster genes, and 53 hyperplasia-related genes known to be essential for adipogenesis[23], were more active in RAD than in GOM or MAD but were comparable with ULB under each nutritional condition (Supplementary Fig. 4d–f). Importantly, mitochondrial biogenesis in ATs is enhanced for hyperplasia but weakened in hypertrophy as obesity developed[24]. Consistent with that study, we found that 13 mitochondria-encoded genes were generally up-regulated in ULB and SAT-like RAD after weight gain, but no such pattern was evident for GOM and MAD (Supplementary Fig. 4g). The compartment status of 36 paralogous homeobox (*HOX*) TFs (the major regulators of animal morphogenesis and development) in RAD was more similar to that in SAT, and was clearly distinguishable from their profiles in GOM and MAD despite their shared physiological location (Supplementary Fig. 4h). These results highlight the well-characterized differences between VATs and SATs, particularly in their respective metabolically harmful or protective roles[17] (Supplementary Fig. 7).

Using the distance between samples in the t-SNE as a metric, we found that the average distance between replicates was slightly less than that between treatment groups (Wilcoxon rank-sum test, $p < 1.96 \times 10^{-9}$), and markedly less than that between pairwise ATs (Wilcoxon rank-sum test, $p < 2.2 \times 10^{-16}$) (Supplementary Fig. 8), suggesting considerable intra-group heterogeneity.

### Population dynamics of compartmentalization and TAD boundaries between ATs

To quantify the intra-population variability in compartmental status and TAD boundaries (Supplementary Note 2 and 3 and Supplementary Figs. 9–17), we examined the consistency of A/B compartments and TAD boundaries within populations, which is the frequency in which a bin was identified in a given compartment or TAD boundary in that population ("Methods") (Supplementary Fig. 18a).

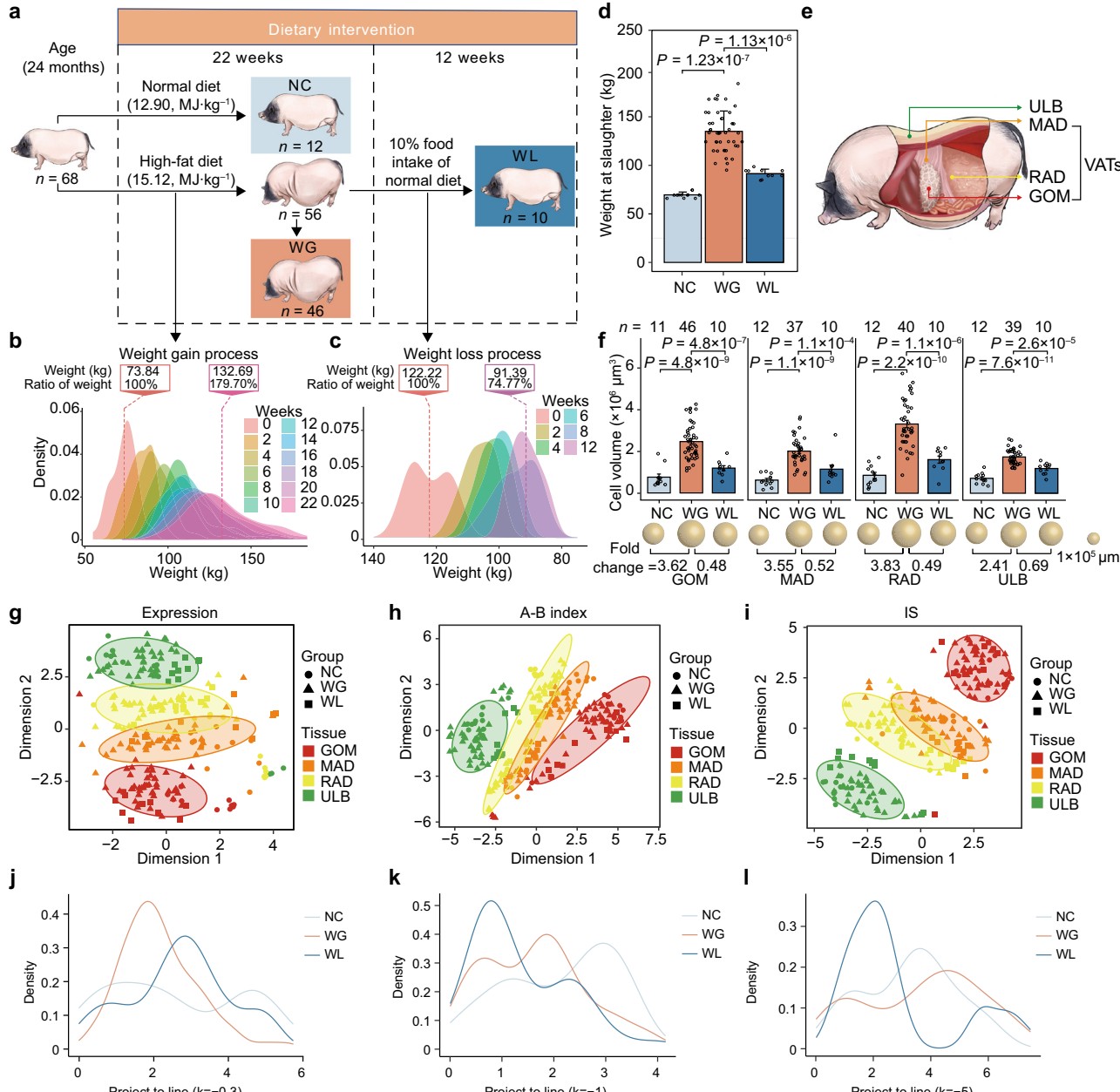

**Fig. 1 | Transcriptomic and chromatin architecture dynamics in distinct ATs associated with body weight in changes. a** Schematic overview of the experimental design for dietary treatments. **b** Body weight of pigs ($n = 56$) during progressive weight gain over 22 weeks, observed every 2 weeks (12 time points). Ratio of weight: relative to the weight at baseline (0 weeks). **c** Body weight of pigs ($n = 10$) during progressive weight loss over 12 weeks (from 23rd to 34th week), observed every 2 weeks (6 time points). Ratio of weight: relative to the weight at the 22nd week. **d** Histogram of body weight at slaughter for NC, WG, and WL groups. WG weight gain, WL weight loss, NC normal diet. Data are presented as means ± SD (NC, $n = 12$; WG, $n = 46$; WL, $n = 10$). $p$ values were determined by two-sided Wilcoxon rank-sum test. **e** Adipose tissue sources: one SAT (ULB: upper layer of backfat) and three VATs (GOM greater omentum, MAD mesenteric adipose, RAD retroperitoneal adipose). SAT subcutaneous adipose tissue, VAT visceral adipose tissue.

**f** Histogram of adipocyte volumes during weight gain or loss for each adipose depot (top). Spheres show relative adipocyte volume, with the scale shown on the right (middle). Fold-changes in adipocyte volume in WG relative to NC (left) and in WL relative to WG (right) (bottom). Data are presented as mean values ± SD. Statistical significance was determined using a one-sided Wilcoxon rank-sum test. **g**–**i** Comparison of variation in gene transcription (**g**), AB compartment (**h**), and IS (**i**) between adipose depots and between groups. t-distributed stochastic neighbor embedding (t-SNE) clustering of samples. In t-SNE plots, ellipses indicate AT samples with similar profiles, constructed at a probability of 0.85. **j**–**l** Proportional distribution of projection distance for t-SNE plots of gene expression in **g** (**j**), A-B index in **h** (**k**), and IS in **i** (**l**) between each dot and a given line ($y = kx$, $k = -0.3, -1, -5$ for gene expression, A-B index and IS, respectively) across groups. Source data for (**b**–**d**, **f**–**l**) are provided as a Source Data file.

The A/B compartments were roughly classified into three categories based on their consistency: high (>70%), medium (30–70%), and low (<30%). This analysis found that most A/B compartments were invariable within replicates (Supplementary Fig. 11a). Moreover, compartmental status remained generally unchanged across ATs and treatments/groups, with changes (i.e.,

frequency of active compartment A status changed from high to low, or reversely) observed in an average of 1.31% (1489 bins of 20-kb length) and 0.29% (327) of bins, respectively (Fig. 2a, b and Supplementary Data 3).

The common active compartment regions across ATs were more prone to enrichment with genes, high GC content, housekeeping

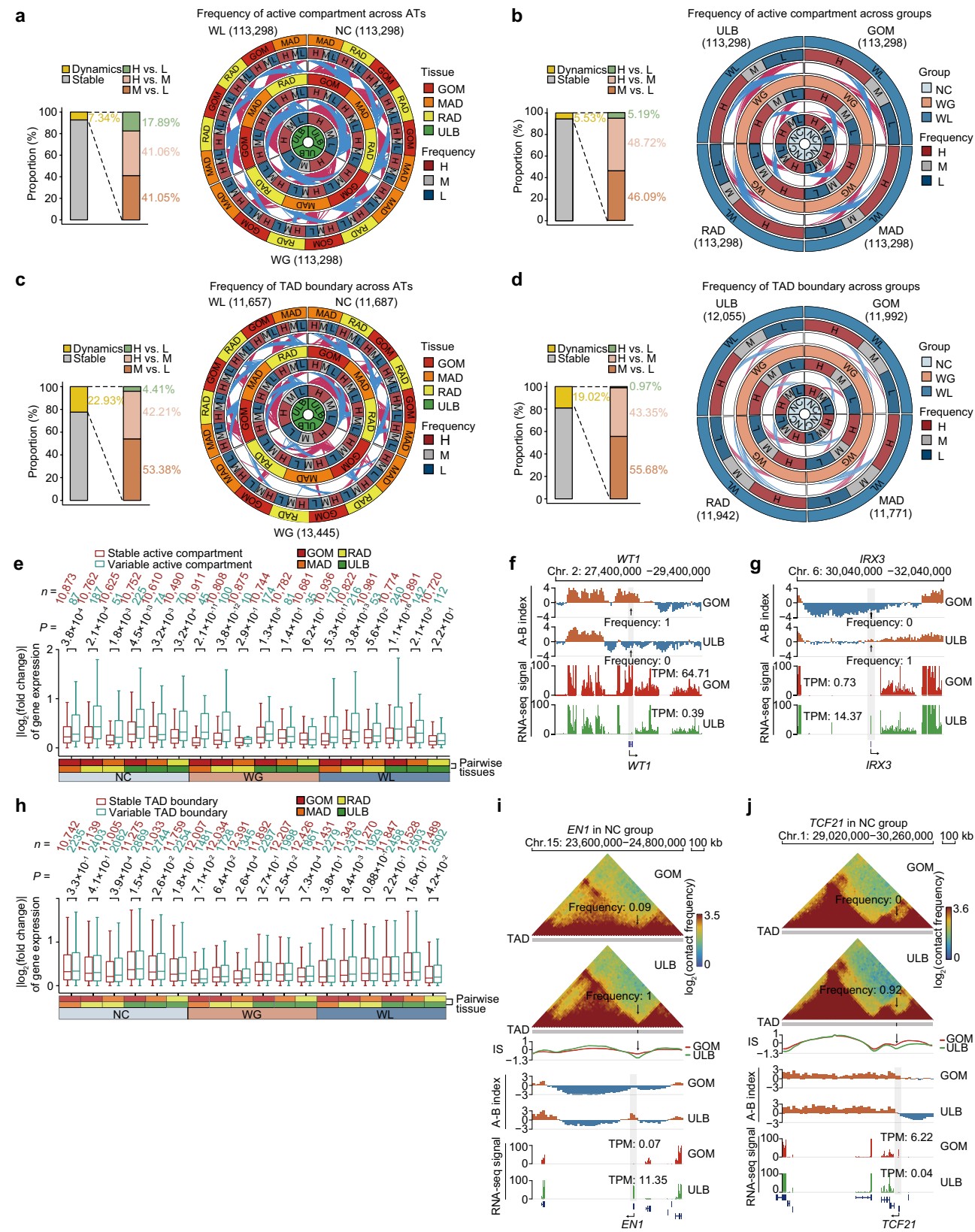

**a** Frequency of active compartment across ATs

**b** Frequency of active compartment across groups

**c** Frequency of TAD boundary across ATs

**d** Frequency of TAD boundary across groups

**e**

**f** *WT1* Chr. 2: 27,400,000 −29,400,000

**g** *IRX3* Chr. 6: 30,040,000 −32,040,000

**h**

**i** *EN1* in NC group Chr.15: 23,600,000−24,800,000

**j** *TCF21* in NC group Chr.1: 29,020,000−30,260,000

genes, short interspersed nuclear elements (SINEs), and evolutionarily conserved sequence but depleted for long terminal repeat (LTRs)[25] (Supplementary Fig. 16). In contrast, genes in compartments that shifted between ATs or conditions exhibited higher changes in expression than those within common bins (Fig. 2e and Supplementary Note 2). For example, in the NC group, the compartmental status

around the VAT-specific developmental regulator *WT1*[26] was commonly active in GOM (frequency = 100%) with elevated expression (TPM = 64.71), but inactive in ULBs (frequency = 0) and weakly expressed (TPM = 0.39) (Fig. 2f). In contrast, the compartmental status around the adipose browning and metabolic activity regulator *IRX3*[27] was commonly active in ULBs (frequency = 100%) with elevated

**Fig. 2 | Population-level dynamics of compartmentalization and TAD boundaries in distinct ATs across weight gain and loss treatment groups.**
**a, b** Differences in the frequency of A/B compartments in pairwise comparisons of ATs and treatments. Proportion plot showing stability and variability of compartment shifts across different ATs (**a**) and groups/treatments (**b**). Circos plots highlight changes in compartment status between pairwise ATs in each treatment group (**a**) or between treatment groups in each AT (**b**). Red or blue vectors indicate changes in frequency from low-to-high or high-to-low, respectively, in pairwise comparisons between ATs in adjacent rings; vector thickness is proportional to frequency of the changed compartment. **c, d** Differences in frequency of TAD boundaries in pairwise comparisons of ATs and treatments. Proportion plot showing the stability and variability of TAD boundary shifts across different ATs (**c**) and groups/treatments (**d**). Circos plots highlight changes in TAD boundary frequency. **e** Box plots of changes in expression level for genes embedded in stable active compartment regions (red) vs. those that changed (green) between pairwise

ATs. **f, g** Representative changes in A/B compartment status between ATs. Compartment status (A/B) across all individuals in the NC group is shown for the typical VAT-active gene *WT1* and SAT-active gene *IRX3*. Expression levels (TPM) were also plotted. **h** Box plots showing changes in expression levels of genes embedded in stable TAD boundary regions (red) vs. those that changed (green) between pairwise ATs. In the boxplot of (**e**) and (**h**), the internal line indicates the median, box limits indicate the 25th and 75th quartiles, and whiskers extend to 1.5 × IQR from the quartiles. The gene number in each category is listed above each box. Statistical significance was determined by a two-sided Wilcoxon rank-sum test.
**i, j** Representative TAD boundaries shift between ATs. Chromatin interaction heat maps are shown for typical replicates of GOM and ULB in the NC group. The black arrow indicates the position of the shifted boundary. Representative embedded genes include *EN1* (**i**) and *TCF21* (**j**), which are highly expressed in ULB and GOM, respectively. Source data are provided as a Source Data file.

expression (TPM = 14.37), but inactive in GOMs (frequency = 0) and weakly expressed (TPM = 0.73) (Fig. 2g).

Similarly, TAD boundaries could also be classified as either high (>70%), medium (30–70%), or low (<30%) frequency based on their consistency (Supplementary Fig. 18b), a minority of which did not vary between replicates. Moreover, TAD consistency was comparable across ATs and nutrition groups, only switching 1.01% and 0.18% TAD (i.e., frequency of TAD boundary changed from high to low, or reversely), respectively (Fig. 2c, d and Supplementary Data 3).

Common TAD boundaries were more prone to enrichment with CCCTC-binding factor (CTCF) binding site motifs, transposable elements (TEs), especially recently inserted SINEs with low sequence divergence and housekeeping genes, and showed markedly stronger insulation than variable ones between ATs[28] (Supplementary Fig. 18c, d and Supplementary Note 3). Consequently, genes near shifted TAD boundaries (i.e., within ±100 kb of a boundary) in some of the comparisons showed slightly higher (though not significant) changes in expression than those near the common boundaries[29] (Fig. 2h). The influence of shifted boundaries on gene expression was relatively milder than that observed in A/B compartments. These differences in expression could potentially reflect the rewiring of the distal PEI networks caused by changes in the chromatin insulation status at these loci. In one scenario, increased chromatin insulation appeared to block non-specific PEIs, thereby increasing contact frequency with their endogenous distal regulators. For instance, the key adipogenic developmental gene *EN1*[30] is adjacent to a common boundary in ULBs, while this boundary is absent in GOMs. Likewise, *EN1* expression in ULB (TPM = 11.35) is higher than in GOM (TPM = 0.07) (adjusted $p = 2.46 \times 10^{-13}$) (Fig. 2i).

Another scenario might be that increased in chromatin insulation status prevent regulatory contact with external enhancers. For example, there is a common boundary around the pro-inflammatory marker *TCF21*[31], which is weakly expressed in ULB (TPM = 0.04). However, this boundary is absent in GOM under NC conditions, and *TCF21* expression is largely increased (TPM = 6.22; adjusted $p = 2.12 \times 10^{-11}$; Fig. 2j). These two examples highlight the presumably distinct roles of TADs in facilitating or constraining interactions between gene promoters and regulatory elements[28,32,33]. Further efforts are required to systematically identify the functional and mechanistic roles of TADs across genomic contexts when governing transcription.

## Rewiring of PEIs and associated transcriptional changes and response to body weight changes in different ATs

To investigate how spatial rewiring of regulatory circuitry could affect the dynamics of transcriptional programs in ATs during WGWL, we used the PSYCHIC algorithm[34] to compile a 5-kb resolution, genome-wide catalog of PEIs across four ATs under three nutritional conditions (Fig. 3a–d and Supplementary Figs. 19 and 20a). This analysis identified a comparable number of PEIs in each of the 12 groups, with an average

of 42,736 enhancers assigned to 10,602 promoters and a median bridging distance of ~131 kb (Fig. 3a). PEIs were preferentially located within TADs (66.5%) ($p < 2.2 \times 10^{-16}$, $\chi^2$ test) or CTCF-mediated loops (49.28%) ($p < 2.2 \times 10^{-16}$, $\chi^2$ test) (see Supplementary Note 4 for details) (Supplementary Figs. 21–25 and Supplementary Data 4).

Supporting the additive effects of multiple enhancers on target gene transcription, genes engaged in physical contact with multi-enhancers (~49.82% genes, TPM = 10.39) showed higher transcription levels than those engaged with a single enhancer (~18.02% genes, TPM = 5.37) or no enhancer (~45.13% genes, TPM = 3.30) interactions (Supplementary Fig. 20c). We found that the chromatin interactome of PEIs was more consistent across the three treatment groups for a given AT than among different ATs within a single treatment group (Supplementary Fig. 20a, b).

To better understand how this extensive PEI rewiring could contribute to transcriptomic divergence, we calculated a regulatory potential score (RPS) for each promoter (Fig. 3b) and further measured enhancer activities by analyzing the distribution of H3K27 chromatin acetylation marks (H3K27ac) (Supplementary Figs. 20d and 26). In total, ~170 genes with covariation between RPS and gene expression were identified. Of these, genes with higher RPS values (FC [fold change] >1.5, Δ > 2) were generally up-regulated ($\log_2$ FC > 1, FDR < 0.05) between ATs under the same nutritional conditions (Supplementary Fig. 27a).

Functional enrichment analysis[35] showed that these VAT-specific genes were primarily involved in responses to inflammation and immunity (e.g., "chemotaxis" and "positive regulation of inflammatory response") (Supplementary Fig. 20e). Typically, VAT-enriched inflammatory markers had more and spatially closer enhancers (and thus higher RPS) and generally had enhancers with more intensive activity (typically, super-enhancers) in VATs than in SAT. This included the monocyte activation and inflammatory response regulator *TGM2*, which is required to clear large, lipid-rich apoptotic adipocytes[36] (Fig. 3e and Supplementary Figs. 28a and 29a), and *CD28*, which activates lymphocyte and T cells to increase adipose inflammation[37] (Supplementary Figs. 20g, 28b and 29b), and *CXCR4, CCR2, IL18,* and *SUCNR1* (Supplementary Data 5). In contrast with VATs, this enriched gene set in ULB was primarily related to lipid metabolism (e.g., "regulation of lipolysis in adipocytes" and "lipid storage") (Supplementary Fig. 20e), including two TFs necessary for adipogenesis[38] (*PPARG* and *CEBPA*) (Fig. 3f and Supplementary Figs. 20h, 28c, d and 29c, d), and *COL6A3, GCG, GPAM, IGFBP5, MAP4K4,* and *SFRP4* (Supplementary Data 5). Remarkably, we detected enhanced regulatory circuitry for nine *HOXD* TFs (sequential gene clusters with enhancer-rich regulatory landscape) in ULB and RAD compared with their network circuitry in GOM and MAD, supporting their active compartment status and increased expression in SAT (Fig. 3g and Supplementary Fig. 30).

We identified ~105 genes with covariation between RPS and gene expression for a given AT between the groups (Supplementary

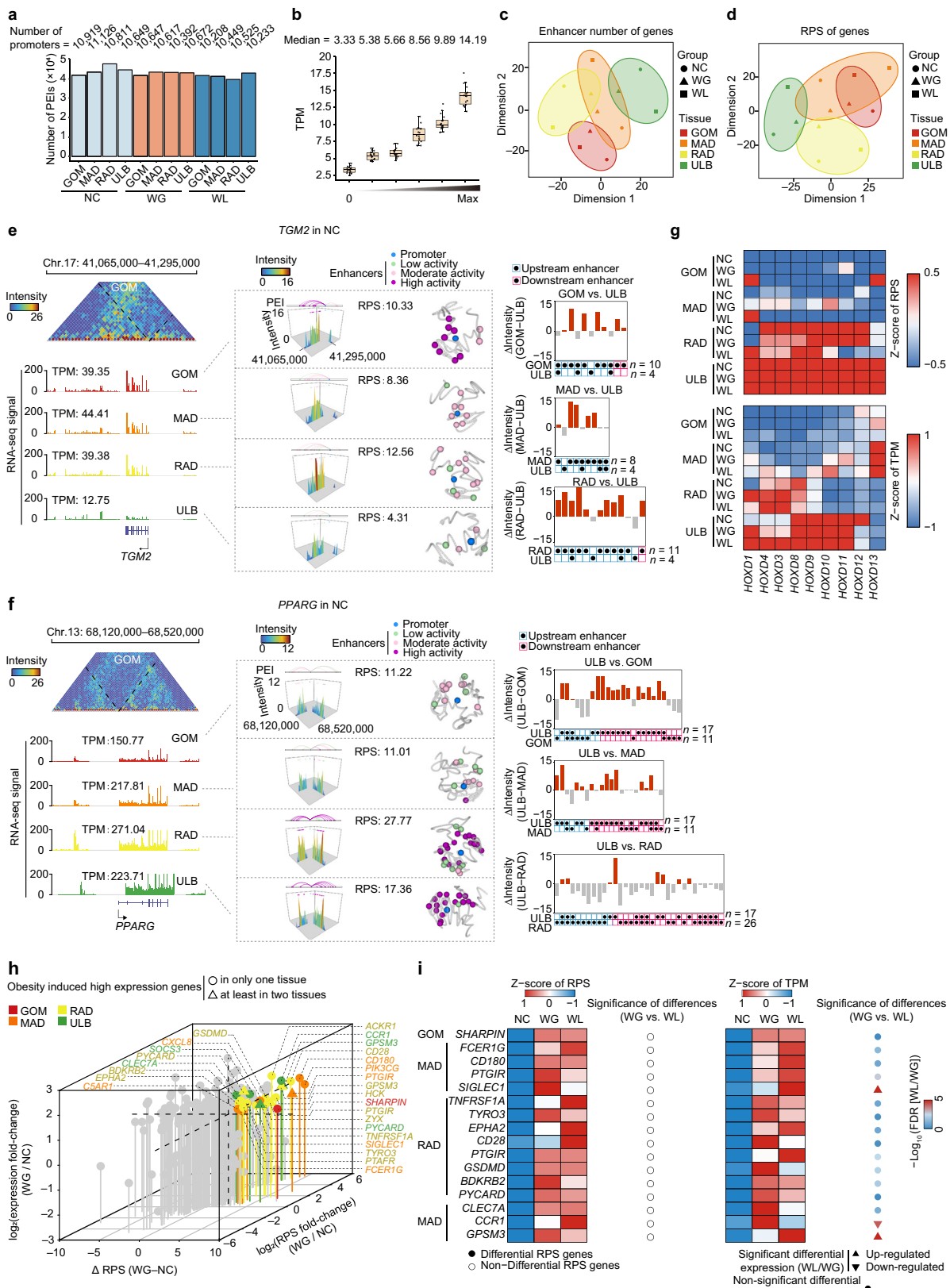

Fig. 27b). Genes with enhanced RPS and expression in the WG group compared to the NC group were primarily associated with inflammation and immune response (e.g., "regulation of chemotaxis" and "leukocyte migration") and fibrosis (e.g., "ECM-receptor interaction") (Supplementary Fig. 20f), supporting the elevated inflammation in obesogenic ATs. Previous studies in humans and mice proposed the phenomenon of "obesogenic memory", in which a long-term increase in AT inflammation and insulin resistance persisted even after weight loss[39]. Consistent with this hypothesis, five markers of "obesogenic memory" in mice (*TNF*, *IL6*, *IL10*, *CCL2*, and *CCL3*)[39] retained high expression and similar pattern of interactions in promoter-centered regions (and persistent AT inflammation) following weight

**Fig. 3 | Rewiring of PEIs with transcriptional changes between distinct ATs and the dynamic during different nutritional conditions. a** The number of PEIs in each AT across groups. The numbers of genes/promoters are indicated above each bar. **b** Positive correlation between gene expression and RPS. Genes with RPS > 0 were divided equally into five percentiles. In the boxplot, the internal line indicates the median, the box limits indicate the upper and lower quartiles and the whiskers extend to 1.5 IQR from the quartiles ($n = 18$). RPS: a regulatory potential score for each gene. **c** t-SNE plots of the number of enhancers that interact with each promoter. **d** t-SNE plots of RPS of genes. The divergence in PRS profiling highlights that differences in features are more pronounced among ATs than among groups. **e** Schematic representation of PEIs for the typical pro-inflammatory gene *TGM2*, which was abundantly expressed in VATs of the NC group. (left) Promoter-centered interactions and expression levels for gene examples across four ATs. (middle left) Interaction metaplots of promoter-centered regions across four ATs. (middle right) 3D structural models of the corresponding genomic regions. (right) Difference in PEI intensity between pairwise AT comparisons. **f** Schematic representation of PEIs for the typical adipogenesis gene *PPARG*, which is abundantly expressed in SATs of the NC group. **g** Heatmaps of RPS values (top) and expression level (bottom) of *HOXD* genes across ATs. **h** Changes in expression and RPS values of 159 known inflammatory genes in each AT between the NC and WG groups. Twenty-three inflammation-related up-regulated genes in each AT of the WG group compared to NC are shown. **i** Heatmaps of RPS (left) and expression (right) patterns of 15 inflammation-related genes that are highly expressed in the WG group compared to the NC group and remained stable in the WL group. Differential RPS genes: genes with changes in RPS FC [fold change] >1.5, |Δ| > 2; Otherwise, Non-differential RPS genes. Source data are provided as a Source Data file.

loss (compared to that in WG group) in our pig model (Supplementary Fig. 31).

Of 23 known inflammatory genes with concordantly enhanced RPS and expression during weight gain, 15 (~65.22%) sustained higher RPS and expression after weight loss (i.e., WL vs. WG groups) (Fig. 3h, i). In particular, the immune and inflammatory response[40] regulator *SHARPIN*, the TLR4-independent AT inflammatory factor *CD180*, the inflammatory cytokine activator *TNFRSF1A*, and the macrophage infiltration and activation marker[41] *CLEC7A* in GOM, MAD, RAD, and ULB, respectively (Supplementary Fig. 32). These results strongly suggested that "obesogenic memory" affects a proportion of inflammatory genes in ATs, which retain their inflammatory state despite weight loss[39]. Further studies are needed to better understand the stored mechanisms of "obesogenic memory".

## Evolutionary divergence of local spatial context in mammalian ATs

Studies using animal models have significantly expanded our understanding of the pathogenesis of obesity and its comorbidities in humans[5]. Nonetheless, the extensive discrepancy between clinical and molecular data in humans and that obtained from other mammalian models prompted us to explore potential evolutionary divergences in chromatin architecture and related influences on transcription that could contribute to the AT-specific biology of mammals[42]. To this end, we explored the evolutionary patterns of local spatial context in SATs (reflected by the IS value, with a higher value corresponding to more open architecture [compartment A status], and thus higher transcriptional activity, Supplementary Fig. 33). This analysis used 29 in situ Hi-C datasets and their corresponding RNA-seq datasets from pigs, humans and five representative mammalian models (including a rodent [mouse], a lagomorph [rabbit], two carnivores [dog and cat], and an artiodactylid [sheep]). Of these, 20 were publicly available from our recent work (Supplementary Data 6). Using the human genome as a reference, we identified 949.94 Mb (or 30.65%) of homologous regions across seven mammals in the human genome (Supplementary Fig. 34a). As expected, evolutionarily closer species shared greater similarity in their patterns of gene expression and local spatial context (Fig. 4a–c) (see "Methods" for details).

We identified ~20.88 Mb regions that were more accessible and had increased local interactions in the human genome using the Phylo-HMGP model[43] (Fig. 4d and Supplementary Fig. 34b, c and Supplementary Data 7, 8 and Supplementary Note 5). These regions were significantly enriched by *Alu* elements (Fig. 4d), which represented the most abundant SINEs, were considered to exhibit continuous proliferation activity throughout human evolution[44], and were positively correlated with chromatin interactions (Supplementary Fig. 35). The specific increase in *Alu* contents in those homologous regions with higher, human-specific, local interactions (indicated by high IS) could reflect TE contributions to species-specific local chromatin status and transcription, potentially driving differences in AT biology among mammals.

These human-specific high IS regions exclusively harbored 14 TF motifs (including *SREBF-1* and *-2*, *TCF-3* and *-4*, *HEY2*, and *NFKB2*) compared to other non-conserved regions (Fig. 4e, f). In particular, recognition motifs for the cholesterol uptake and biosynthesis genes *SREBF-1* and *-2* [45], which are closely associated with long-term energy storage and are among the thrifty genes that efficiently use limited energy for AT storage[46], showed profound enrichment in human-specific high IS regions but were absent in other non-conserved regions. Moreover, the enrichment pattern of these two TF motifs was also evident when compared to homologous regions of other species (*SREBF1* enrichment score: human, 1.16 vs. average 1.01 in other species; *SREBF2*: 1.17 vs. 1.00) (Supplementary Fig. 34e). This specifically active chromatin status related to de novo lipid biosynthesis in humans supports a hypothetical evolutionary origin of obesity in which humans evolved a "thrifty mode of fuel utilization" to store excess nutrients as ATs and prepare for cyclical episodes of famine and surplus after the advent of farming ~10,000 years ago[45,47]. This example suggests a link between species-specific changes in local spatial genomic context and cis-regulatory elements, such as unique TF binding sites.

## Comparison of enhancer regulatory circuitry to identify orthologs in human and pig AT genomes

The above findings in a porcine model supported obesogenic observations in humans and rodents[48,49], leading us to further explore its use as a biomedical model for humans. We next examined whether the biologically meaningful effects of a gene's regulatory target in pigs could be extrapolated to humans by systematically evaluating divergence in PEI organization between human and pig genomes across the four ATs in the normal condition group. By combining publicly available Hi-C and RNA-seq datasets of three human SAT samples, we obtained a Hi-C dataset containing 7.46 billion total validly aligned contacts (reaching a maximum resolution of ~2 kb by merging intrachromosomal contacts of 5–7 biological replicates for each AT) and 25 total RNA-seq datasets (~12.20 Gb of high-quality sequences per sample) for human ATs (Supplementary Data 9). We identified 34,638 enhancers assigned to 11,861 promoters for each human AT, with a median bridging size of 161.25 kb (Fig. 5a, b and Supplementary Fig. 36a–e). The global pattern of higher divergence between VATs and SAT compared to divergence within VATs was obscured by the relatively greater variation among human samples. However, the relatively higher divergence between inflammatory GOM and metabolic SAT (i.e., ASA) was still evident in RPS analysis and transcriptomic profiles, as well as in our analysis of local spatial context (reflected by A-B index and IS values). Moreover, we also observed that visceral RAD samples were closer to SAT, distinguishable from the congeneric GOM and MAD, similar to our findings of ATs in a pig model (Supplementary Fig. 37).

Evolutionary pressures during speciation and adaptation generally lead to regulatory innovations by allowing sequence variation in enhancers to subtly alter transcription at existing or newly adopted

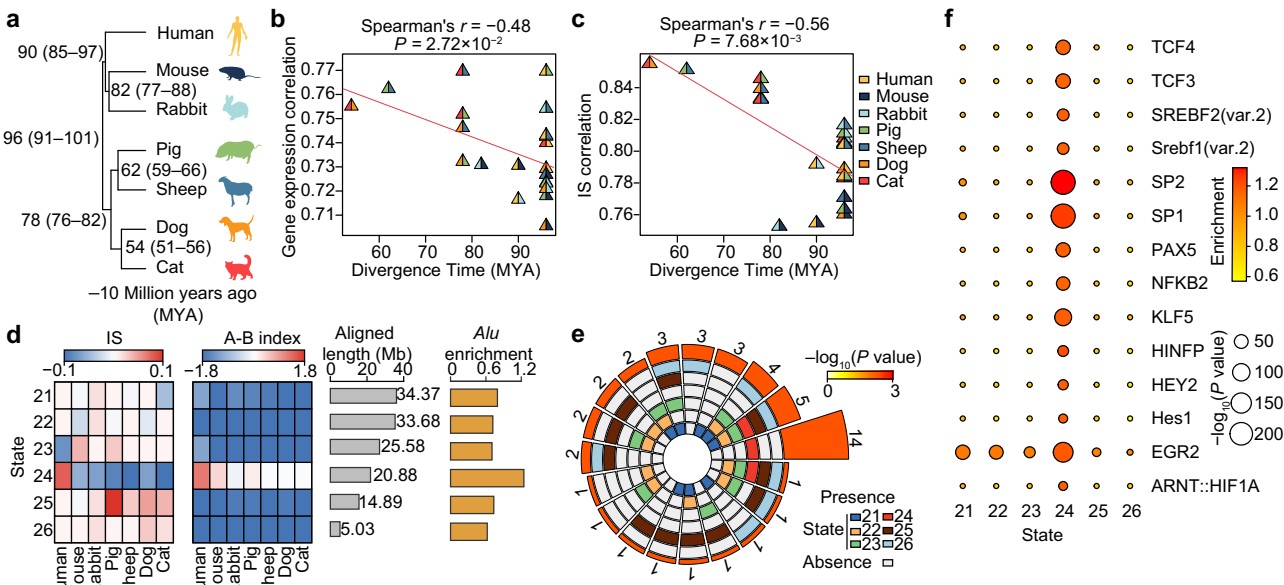

**Fig. 4 | Evolutionary divergence of local spatial context in ATs across seven mammalian species. a** Divergence times (million years ago, MYA) and phylogenetic tree topology of seven representative mammalian species retrieved from the TimeTree database (http://www.timetree.org/). The number in the parentheses indicates the range of divergence times. **b** Plot of Spearman's *r* values showing the relationship (*r* = −0.481) between gene expression levels of 1-1 orthologs (*n* = 8020) and the divergence time for each pair of mammalian species. The statistical significance of the two-sided *p* value was calculated using hypothesis testing. **c** Plot of Spearman's *r* values showing the relationship (*r* = −0.564) between the IS index of all bins (94,994 10-kb bins) and the divergence time for each pair of mammalian species. The statistical significance of the two-sided *p* value was calculated using hypothesis testing. Evolutionary distance is positively correlated with divergence in gene expression levels and with IS divergence across species. **d** Distribution pattern

of IS levels, A/B compartment status, and *Alu* element for non-conserved states (state 21–26, see Supplementary Fig. 34 for details). Note the enrichment for SINE-*Alu* elements in human-specific high IS state 24. **e** Circular plot illustrating the overlap/intersection of transcription factor motifs enriched across non-conserved states. The six tracks in the middle represent the six states, with individual blocks showing the "presence" (colored) or "absence" (white) of the state in each intersection. The height of the bars in the outer layer is proportional to the intersection size. Numbers above each bar indicate overlapping TFs for each state. The color intensity of the bars represents the significance of the intersections, as denoted by its *p* value (χ² test). **f** Enrichment of 14 TFs that were specifically enriched in state 24. The statistical significance of the *p* value was calculated using χ² testing. Source data for (**b**–**f**) are provided as a Source Data file.

target genes, more often than through changes in protein-coding sequences[50]. By mapping the contacts of each AT in pigs (the "query" species) to the human genome (the "reference" species) (Fig. 5c), we were able to statistically compare inter-species differences in the contact frequency of most PEIs in the human genome (~10,781 of 11,861, or ~90.89% of homologous comparable promoters) (Supplementary Fig. 36f). We also assessed enhancer conservation (~31,341 per AT) by comparing their sequence divergence and functional usage (i.e., harboring potential acting as a *cis*-regulatory element) (Fig. 5c) (see "Methods" for details).

Notably, human enhancers that interacted with 8989 single-copy orthologs (shared by humans and pigs) were remarkably conserved in sequence (83.56% of all enhancers) and usage (60.96% of all sequence-conserved enhancers). This was higher than those interacting with multi-copy orthologs (sequences: 48.85%; usage: 55.25%) or human-specific genes (sequences: 59.76%; usage: 57.35%) (Fig. 5d). Similar results were observed when mapping human contacts to the pig genome (i.e., swapping "query" and "reference" species) (Supplementary Fig. 38). For example, the adipogenic TFs *CEBPB*[51] *and PPARG*[23], homeostasis regulator *IGFBP5*[52], inflammatory signal factor *IL6ST*[53,54], and the mediator of oxygen and nutrient exchange during fat mass expansion a *VEGFA*[55], each have over 20 enhancers in human AT genomes, most of which are conserved in both sequence and usage between humans and pigs (Fig. 5e and Supplementary Data 10 and 11).

Enhancers of single-copy orthologs that were differentially expressed between humans and pigs harbored enhancers with more rapidly evolving sequence (72.29%) and usage (48.86%) than those with comparable inter-species expression (sequence: 84.42%, usage: 61.75%) (Fig. 5f and Supplementary Data 12). These differentially

expressed genes were biologically important to core AT functions such as lipid localization and insulin signaling (Supplementary Data 13). For instance, *LMO3*, a human-specific regulator known to modulate the activity of the adipogenic master regulator *PPARγ*[56], was more highly expressed in human ATs (TPM = 35.35) compared to pigs (1.25). This gene also harbored more species-specific enhancers in the RAD of humans than in pigs (2 human-specific enhancers vs. 1 enhancer in pigs). *UCP2*, which is involved in energy expenditure[57], interacted with ~5 specific-usage enhancers and was highly expressed in humans (TPM = 38.83), but showed relatively low expression (TPM = 4.75) and had no regulatory enhancers in pigs (Supplementary Data 12).

*FMO1* is a regulator of energy homeostasis, the loss of which leads to triglyceride depletion in white ATs[58]. *FMO1* was abundantly expressed in the metabolic tissue (especially liver) of pigs and rabbits but was not expressed in adult humans[59]. Consistent with this previous study, we found that *FMO1* interacted with 16 enhancers in porcine SAT (four were specific to the pig genome, nine had conserved sequence, and three were conserved in usage), and only contains two enhancers in human SAT (one of which is human-specific and the other had conserved sequence). This gene was thus highly expressed in pig (TPM = 86.50) compared to its expression in human (TPM = 1.73) (Supplementary Data 12).

In addition to the above examples, the triacylglycerol and glycerophospholipid biosynthetic pathway gene *AGPAT2*[60], which is essential for postnatal development and maintaining both white and brown ATs[61,62] was significantly more highly expressed in human ATs (TPM = 316.46 on average) compared to pigs (23.60) (Supplementary Data 12). Notably, only one human-specific enhancer of *AGPAT2* was identified. In addition to these functions, AGPAT2-lacking adipocytes

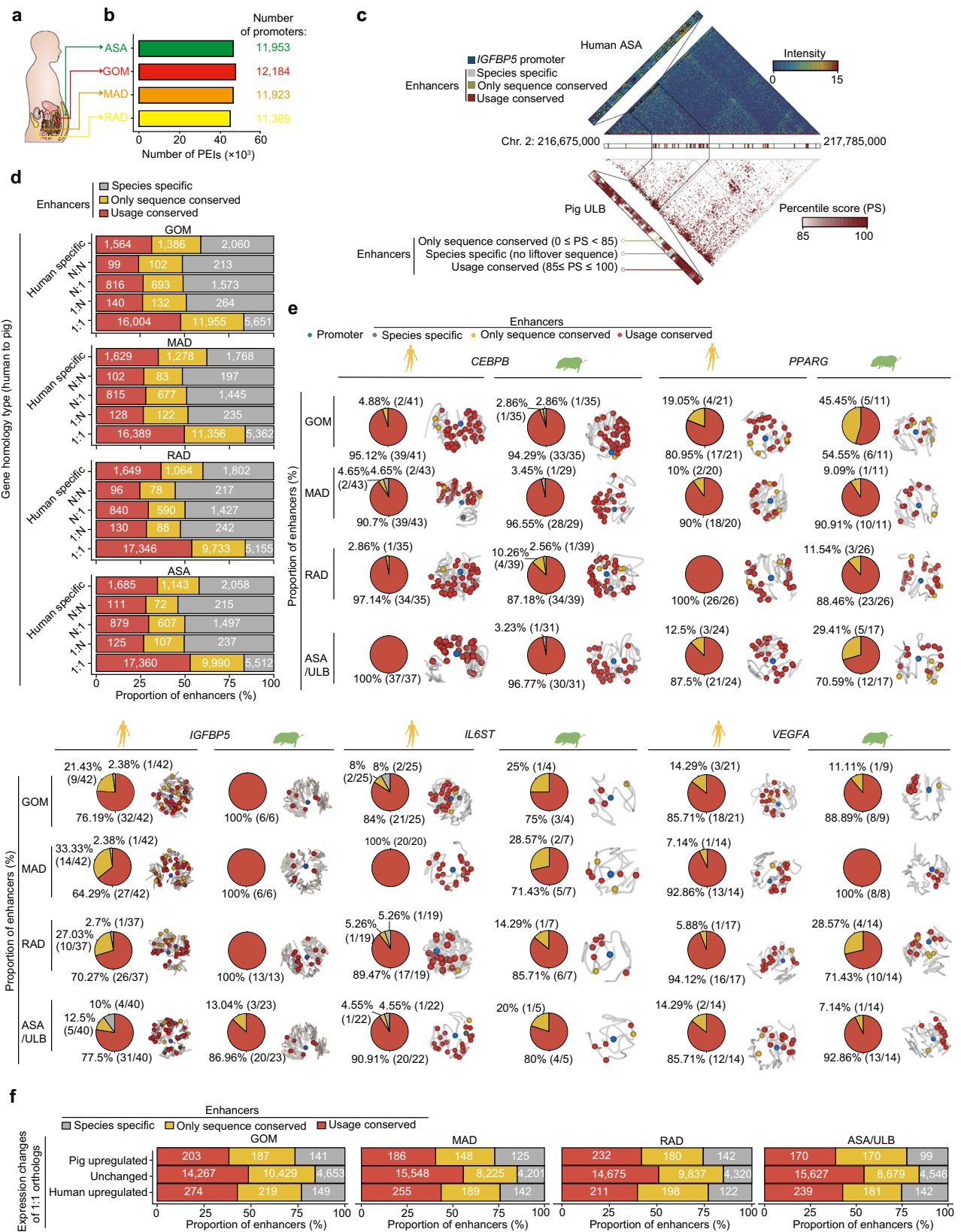

show decreased caveolae[63] and are reportedly involved in the adaptive fat mechanism, thus promoting cold tolerance in the sable[64]. These results align with accumulating evidence indicating that evolutionary disruption of the brown adipocyte marker *UCP1* could have caused the loss of brown AT in pigs (Supplementary Fig. 36g) and may be the leading cause of cold stress-induced neonatal mortality and increased

adiposity in pigs[65–67]. Additionally, species-specific enhancers were generally contacted by human-specific genes, such as *MTLN*, which is involved in lipolysis- and mitochondrial β-oxidation-mediated triglyceride clearance from adipocytes[68], as well as the neuron apoptosis inhibitory protein and adipocyte differentiation regulator *NAIP* [69] (Supplementary Fig. 36g).

**Fig. 5 | Inter-species conservation of human enhancers connected to orthologs between humans and pigs in each AT. a** Schematic of anatomic locations of AT depots in humans. **b** The number of PEIs in each AT. The number of genes/promoters are indicated above each bar. **c** Example evaluation of enhancer conservation in human ATs performed by mapping the contacts of PEIs from humans to pigs. In this case, PEIs of the *IGFBP5* gene in human ASA are shown. According to the distance-normalized contact frequency (percentile score [PS]) (using C-intersecture software, see "Methods" for details), three types of enhancer conservation, including only sequence conserved (yellow), usage conserved (red), and sequence-specific (gray), are indicated in the magnified area. **d** Percent distribution of enhancers that are conserved in both sequence and usage for different types of orthologs between humans and pigs in each AT. The number of enhancers is shown in each bar. **e** Typical 1-1 ortholog examples (*PPARG*, *CEBPB*, *IGFBP5*, *IL6ST*, and *VEGFA*) contain enhancers that are highly conserved between humans and pigs. The conservation of enhancers for each gene is shown using humans (left) and pigs (right) as "reference" species, respectively. **f** Percentage distribution of enhancer conservation for 1-1 orthologs with significantly different changes in normalized expression between humans and pigs (5% confidence intervals) in each AT. The number of enhancers is shown in each bar. Source data for (**b**–**f**) are provided as a Source Data file.

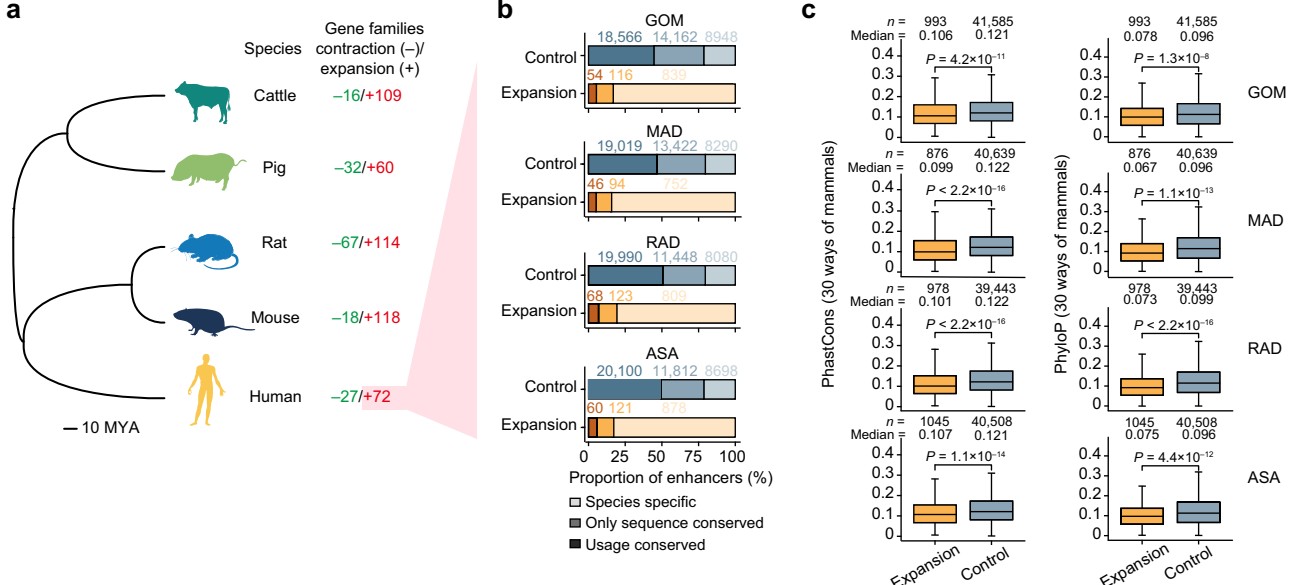

**Fig. 6 | Inter-species conservation of human enhancers of rapidly evolving duplicated genes in each AT. a** Phylogenetic relationship of five representative mammalian species. The number of gene families showing significant expansion or contraction is displayed in red and green, respectively. **b** Inter-species conservation of human enhancers for gene families showing significant expansion compared to that for other families (as control). **c** Boxplots showing evolutionary conservation (inferred by PhastCons and PhyloP values) for enhancers of expanded genes and other genes. In the boxplot, the internal line indicates the median, the box limits indicate the upper and lower quartiles and the whiskers extend to 1.5 IQR from the quartiles. The numerical value above each bar indicates number of genes within this group. *p* values determined by one-sided Wilcoxon rank-sum test. Source for (**b**, **c**) data are provided as a Source Data file.

Altogether, these results illustrate the evolutionary similarity between regulatory circuitry that can determine the obesity phenotype and the emergence of other genes harboring non-conserved elements that could lead to species-specific changes in AT biology[70].

**Gene duplication accompanied by rapidly evolving enhancers**
Gene duplication can be a primary contributor to the acquisition of new functions and physiology[71]. Compared with between-species homologs, within-species homologs that arise through a duplication event generally exhibit higher similarity in both their expression and chromatin architecture (local spatial context and RPS). This finding is consistent with accumulating evidence apparently contradicting the "ortholog conjecture"[72] (Supplementary Fig. 39). Compared to enhancers that interacted with stable copy number gene families, we found that enhancers that interacted with expanding, human-specific, gene families ($n$ = 72) evolved more rapidly, as indicated by lower nucleotide sequence conservation (phastCons values: 0.103 vs. 0.117, $p$ = 0.0001 and phyloP values: 0.073 vs. 0.085, $p$ = 0.0001, Wilcoxon rank-sum test) and less conservation between human and pig sequences (16.91% vs. 59.83%) and usage (34.46% vs. 57.49%) (Fig. 6).

For instance, the *CEACAM* gene family shows substantial expansion in humans, with 18 members in humans and only five in pigs. This family is essential for maintaining insulin sensitivity in ATs by mediating insulin transport through the endothelial cell barrier[73,74]. Of the

18 human *CEACAM* genes, only eight promoters had comparable homologs in pigs, with most enhancers specific to the human genome. For example, 1 of 2 *CACAM1* enhancers in GOM, 2 of 2 in MAD, 2 of 3 in RAD, and 2 of 3 enhancers in ULB were not found in the pig genome (Supplementary Data 14). Additionally, the human-specific and expanding *NBPF* gene family, which is associated with brain size and cognition[75,76], has 15 members in humans and six in pigs. Of the 15 human *NBPF* genes, only nine promoters had comparable homologs in pigs. Most enhancers for nine *NBPF* genes (e.g., ~6 of 6 enhancers in GOM) were specific to the human genome (Supplementary Data 14).

We also found cases of gene duplication that could potentially underpin human-specific physiology. The water and glycerol transport-related *AQP7* family (2 members in humans compared to 1 in pigs and three other mammals) is responsible for glycerol use (i.e., energy) from adipocytes for energy[77,78]. This family was hypothesized to contribute to human-specific metabolic adaptations in endurance running and thermoregulation by increasing sweating[75]. The two copies (*AQP7* and *AL845331.2*) present in humans have comparable homologous promoters in pigs. Across the four ATs, only two enhancers targeting *AQP7* in GOM could be identified, one of which was specific to the human genome, leaving one with a conserved sequence comparable to that of pigs. Remarkably, all enhancers targeting *AL845331.2* (ranging from 7 in ULB to 13 in RAD) were specific to the human genome (Supplementary Data 14).

These collective results suggest that evolutionarily conserved genes preferentially interacted with more conserved enhancers. In contrast, rapidly evolving regulatory elements could arise in species-specific events, which could play a crucial role in later adaptive phenotypes of species.

## Discussion

In recent decades, substantial efforts in animal models have significantly increased our understanding of the pathogenesis of obesity and its comorbidities in humans[5]. Although rodent models can provide important mechanistic insights, they have limited predictive value for human therapeutic outcomes. Although previous transcriptomic comparisons among different tissues or cell populations across mammalian models have suggested that evolutionary changes in transcription occur at a higher rate in ATs than in other metabolically active tissues[6], corresponding tissues in evolutionarily closer species appear to have more similar transcriptomic profiles and local spatial context than tissues within the same species (Supplementary Note 5). Supporting the extensive discrepancies in clinical and molecular data between humans and other mammalian models[5], we observed non-trivial divergences in chromatin architecture and corresponding effects on transcription that could contribute to phenotypic, physiological, and functional differences in their respective ATs.

More recently, domestic pigs (and miniature breeds in particular) have increasingly contributed to metabolic research[6]. Moreover, a pig model offers several advantages over other models, including a low genetic variance and distinct phenotypes within and among breeds that are useful for studies requiring homogeneous feeding regimens[79]. Genetically engineered pig lines generated via gene editing and somatic cell nuclear transfer have provided useful models for studying diabetes and dyslipidaemia and can also serve as a potential source for xenotransplantation[80]. In this study, we investigated the different roles of anatomically distinct ATs in the pathogenesis of obesity and its comorbidities in pigs and found that regional differences in SAT and VATs are strikingly similar to those in humans, further supporting their value as an informative model for studying human obesity[6]. Notably, many specific AT populations in humans have no precisely correlating populations in rodents, and vice versa. For example, humans harbor a large VAT mass in the omentum, which is barely present in rodents. Conversely, the large epididymal fat pads found in male mice, which are frequently regarded as representative VAT, are, in fact, absent in human males[81]. Moreover, various animals exhibit anatomical differences in ATs that are related to highly specialized functions[82]. In sheep, but not in other mammals, large amounts of fat can be stored in the tail region as an energy reservoir during migration-associated food scarcity[83]. Alternatively, the activation of energy expenditure pathways (especially in brown ATs) represents one of the most promising areas of therapeutic development for obesity and metabolic diseases[84]. However, neither functional *UCP1*, a key element in BAT-mediated adaptive non-shivering thermogenesis, nor brown adipocytes have yet been identified in pigs[65].

We systematically compared PEI organization in four human ATs with that in pigs to explore whether the biologically meaningful effects of a given gene target in pigs can be extrapolated to humans. Our phylogenomic analyses significantly contributed to the current annotation of regulatory DNA elements (i.e., enhancers) in the pig and human AT genomes. As expected, the DNA sequence variations, especially non-coding SNPs, that are associated with obesity-related traits in humans were also significantly enriched in enhancers in human ATs and in homologous enhancers in the pig genome. This finding confirmed the evolutionary robustness and functionality of these regulatory elements and demonstrated that their high conservation is a regulatory mechanism related to obesity between humans and pigs (Supplementary Fig. 40). Additionally, the DNA

contact maps presented in this work can serve as an instrumental resource for linking obesity-associated variants with target genes, since these variants often appear in non-coding sequences and control transcription by physical contact, and can facilitate the discovery of their functional relevance[85,86]. To facilitate further metabolic investigations with this data, we compiled a publicly available online resource (https://3dgphat.sicau.edu.cn/HPC/front/#/) that integrates gene expression, data related to the chromatin architecture of the four ATs for pig and human genomes, and a browser to comparatively explore pig and human genomes.

In this study, we used a pig model to analyze the multi-scale structural dynamics of the 3D genome and identify the chromatin architectural structure of transcriptional programming associated with progressive metabolic risks accompanying weight gain or weight loss in anatomically distinct adult AT populations. In designing this dietary intervention experiment, we considered the relatively long duration (weight loss over 12 weeks) and the potential time-related effects on the targeted ATs and body weight. To minimize unwanted effects, we employed 2-year-old pigs, approximately corresponding to 30-years-old in humans[87], as representative of fully mature adult stage mammals. Given the relatively constant body weight, fat mass, and phenotypic features of ATs in 2-year-old adult pigs in the absence of nutritional modification, 12-week difference in age between WG and WL animals does not significantly affect phenotypic or genomic functional features of the AT examined in this work. Thus, the currently observed differences/changes in transcription and chromatin architecture between WG and WL group are mostly induced by nutritional intervention. These observations of the intrinsic functional and metabolic differences between SATs and VATs during the development of HFD-induced obesity or dietary restriction-induced weight loss highlight the substantial changes in AT cellular composition that occur during WGWL (Supplementary Fig. 41a). However, our bulk Hi-C data only provide the average features of chromatin architecture in adipose cell populations, though these results largely confirm the phenotype and underlying mechanisms of obesity development in humans and rodents[48,49]. By limiting our comparative transcriptomic, A/B compartment, and RPS analyses to a subset of samples with relatively similar cell composition, we could exclude the influence of variation in cell composition (e.g., greater inflammation-related gene expression due to larger macrophage populations) (Supplementary Figs. 41b-d and 42). We observed that inflammation-related gene activity increased in adipose tissues after weight gain, suggesting that HFD-induced obesity could indeed lead to an elevated inflammatory transcriptional response and/or chromatin rewiring in adipose cells. Single-cell resolution is required to determine the extent to which cellular heterogeneity in ATs contributes to differential signals among chromatin features.

## Methods
### Experimental design and phenotype
**Animals.** All research involving animals was conducted according to Regulations for the Administration of Affairs Concerning Experimental Animals (Ministry of Science and Technology, China, revised in March 2017), and approved by the animal ethical and welfare committee (AEWC) of Sichuan Agricultural University under permit No. DKY-B20161707. A total of 68 2-year-old female adult (physically mature) Bama pigs (an indigenous Chinese miniature pig breed widely used in biomedical studies)[88,89] with an average body weight of 75.00 ± 7.34 kg were used in this study. All 68 pigs were fed well-characterized normal diets, providing 12.90 MJ kg$^{-1}$ of metabolizable energy [ME], 13.46% crude protein, 2% fat, and 5.5% lysine, according to the nutritional requirements outlined by the Feeding Standard of Swine (NY/T 65-2004) and published by the Ministry of Agriculture and Rural Affairs of the People's Republic of China. All animals were acclimated to this normal diet and feeding environment for 1 week.

Fifty-six pigs were switched to a well-characterized high-fat diet (provided 15.12 MJ kg$^{-1}$ ME, 11.26% crude protein, 16.8% fat, and 5% lysine) for 22 weeks, while the remaining 12 pigs continued to receive a normal diet. Animals were fed twice daily on a restricted schedule and dietary dose (3% of body weight monthly).

At the 22nd week of feeding with the high-fat diet, ten pigs were randomly selected for inclusion in the weight loss group. These animals were then restricted to 10% of the daily caloric intake of the normal diet for 12 weeks.

The animals were allowed access to water ad libitum and lived under the same controlled conditions (temperature, 18–22 °C; relative air humidity, 30%–70%). During the experiment, obesity-related phenotypes (e.g., body weight, body length, chest perimeter, neck perimeter, and abdominal circumference) were measured and recorded every 2 weeks.

Animals on the high-fat diet (i.e., WG group) became markedly obese (two-tailed $t$-test, $p < 10^{-5}$) throughout the study, with an overall weight after 22 weeks that was nearly twice that of animals fed the normal diet (i.e., NC group) (WG vs. NC: 134.96 ± 20.77 vs. 69.57 ± 2.92 kg average weight). The ten obese pigs showed a dramatic reduction in body weights following 12 weeks of caloric restriction (i.e., WL group) (average weight for WL group, 91.39 ± 4.66 kg).

**Circulating indicators of metabolism in serum.** Venous blood (50 ml) was collected from each fasted pig immediately before they were euthanized. The whole blood was immediately centrifuged at 1800 × $g$ for 10 min at room temperature, and the resultant sera were stored at −80 °C.

Serum concentrations of total cholesterol (TC), triglycerides (TG), high-density lipoprotein (HDL), low-density lipoprotein (LDL), lactate dehydrogenase (LDH), and glucose (GLU) were individually determined for each pig using a CL-8000 clinical chemical analyzer (Shimadzu) and standard enzymatic procedures.

Consistent with previous studies[15,88], obese pigs showed significant increases in TC, TG, HDL, LDL, and LDH levels, indicating metabolic dysfunction. These indicators were markedly reduced to levels comparable with pigs fed a normal diet after weight loss.

**Tissue collection.** Animals were humanely killed as needed to reduce suffering and were not fed the night before they were killed. In total, each of four adipose tissues (ATs) from different anatomical sites were rapidly and manually separated from each carcass, immediately flash frozen in liquid nitrogen, and stored at −80 °C until RNA and DNA extraction. The four ATs were divided into two groups, including (1) one type of subcutaneous AT (SAT; upper layer of backfat [ULB]) and (2) three types of visceral ATs in the abdominal cavity (VAT; greater omentum [GOM], mesenteric adipose [MAD], and retroperitoneal adipose [RAD]).

**Histology of adipocytes and measurements of adipocyte volume.** For histological examination, all ATs were fixed in 10% neutral buffered formalin solution, embedded in paraffin using a TP1020 semi-enclosed tissue processor (Leica), sliced at a thickness of 6 μm using RM2135 rotary microtome (Leica), and stained with hematoxylin and eosin (H&E). The mean diameter of an adipocyte cell was calculated as the geometric average of the maximum and minimum diameter, and all adipocytes (all cells per field; three sections per sample) were measured for each sample with a TE2000 fluorescence microscope (Nikon) and Image Pro-Plus 7.0 software (Media-Cybernetics). The mean adipocyte volume ($V$) was obtained according to the following formula:

$$V = \frac{\pi}{6} \times \sum_{i=1}^{n} \frac{D_i^3}{n} \qquad (1)$$

where $D_i$ is the diameter of the adipocyte, and $n$ is the number of adipocytes.

**Transcriptional profiling**

**RNA-seq library generation.** Total RNA was extracted using an RNeasy Mini Kit (Qiagen). Samples were depleted for rRNA (Ribo-Zero kit, Epicentre), and libraries were generated with Illumina TruSeq stranded RNA-Seq library kits according to the manufacturer's instructions. All libraries were quantified using a Qubit dsDNA High Sensitivity Assay Kit (Invitrogen) and sequenced with the HiSeq X Ten (Illumina) platform to produce an average of ~11.97 Gb high-quality data for each library.

**RNA data analysis.** All annotated protein-coding genes (PCGs) in the reference genome (Sscrofa11.1, release 102), together with long noncoding RNAs (lncRNA) and transcripts of unknown coding potential (TUCP) transcript annotations from a previously study[6], were used for comprehensive gene annotation and subsequent transcript quantification. The gene symbols for 29 PCGs were manually updated according to the protocols used by a previous study[90] for cases where the gene symbols were not available in the annotation file but were available under the gene description on Ensembl, were updated in a future Ensembl release, or where multiple Ensembl IDs corresponded to a single gene symbol. Paired-end reads were aligned to reference genomes using STAR[91] (version 2.6.0c) with default parameters. Gene-level expression was estimated as transcripts per million (TPM) using the high-speed transcript quantification tool Kallisto[92] (version 0.43.0) with parameters (--bias --rf-stranded). Mapped read counts per gene were extracted using tximport (version 1.6.0) in the R package with default parameters. We considered a PCG to be detected/expressed if it had expression levels greater than 0.5 TPM in at least one sample. For lncRNA and TUCP, we used a cutoff of 0.1 TPM in at least one sample.

Differentially expressed genes (DEGs) were identified by edgeR[93] (default parameters, version 3.30.3) using filtering thresholds of false discovery rate (FDR) < 0.05 and fold change >1.5, and were detected in at least 80% of replicates/samples from at least one group.

**Estimating relative cell-type proportions in adipose tissues using bulk RNA-seq data.** We applied CIBERSORTx[94] (version 1.05, https://cibersortx.stanford.edu) to identify cell proportions using RNA-seq data for all samples. A signature matrix was generated based on transcriptome TPM values of representative genes for three cell types (adipocytes, M1/M2 macrophages, and microvascular endothelial cells [MVEC])[95]. Default parameters were applied. The proportion of adipose tissue was further measured by deconvoluting gene expression levels (in TPM) of bulk RNA-seq data based on the above signature matrix and using S-mode batch effect correction.

**3D genome architecture**

**In situ Hi-C library generation.** Hi-C libraries of each AT were generated according to previously published Hi-C protocols with some minor modifications[96]. Briefly, 1 g AT was pulverized, and 37% formaldehyde was added to produce a final concentration of 4% for chromatin cross-linking. The mixtures were incubated at room temperature (20–25 °C) for 30 min, and glycine was added to produce a final concentration of 0.25 mol/l to quench the formaldehyde. The mixtures were then centrifuged at 1500 × $g$ for 10 min at room temperature, and lysis buffer was added to the upper layer containing adipocytes and homogenized. Homogenate was centrifuged at 5000 × $g$ for adipocyte sediments. Nuclei of formaldehyde-fixed ATs were permeabilized, and DNA was digested with 200 units of *Mbo*I (a 4-cutter restriction enzyme) for 1 h at 37 °C. The restriction fragment overhangs were filled and labeled by biotinylated nucleotides and then ligated in a small volume. After cross-link reversal, ligated DNA was purified and sheared to a length of 300–500 bp, at which point ligation junctions were pulled down with streptavidin beads and prepped for Illumina NovaSeq 6000 sequencing.

**Initial processing of Hi-C data**. Hi-C reads were processed using the Juicer pipeline[97] (version 1.5.6) with default parameters. Sequence data were aligned against the pig reference genome (Sscrofa11.1). Contact reads mapped to sex chromosomes or the mitochondrial genome or associated with low-quality alignments (defined as one or both reads failing to meet a threshold MAPQ ≥ 30) were filtered out. For each autosome, the normalized observed contact matrices were generated using the Knight-Ruiz algorithm (to remove intrinsic biases within the matrix) in the Juicer toolkit with quantile normalization conducted using BNBC (version 1.0.0) (to remove biases between samples), both set to default parameters[98]. The correlation between normalized matrices was calculated using HiCRep[99] (version 1.10.0) with default parameters based on normalized observed contact matrices at 100-kb resolution. We then transformed the normalized observed contact matrices into an observed/expected (O/E) matrix by dividing each normalized observed contact frequency by its corresponding expected contact frequency (calculated as the average observed contact frequency for all loci at a certain distance), using the publicly accessible script, generate.oe.matrix.py (https://github.com/JiamanZhang/Lab_Porcine-Adiposes_paper_codes/tree/main/Lab_OE_matrix).

**A/B compartment determination and analysis**. A/B compartments at 20-kb resolution were identified using both principal component analysis (PCA) and A-B index, as previously described[100]. PCA was performed to generate PC1 vectors for each autosome per sample at 100-kb resolution using the "prcomp" function in R (version 3.6.1) with default parameters. Spearman's coefficient *r* was calculated between PC1 and genomic characteristics, including gene density and GC content, for each autosome using the "cor" function in R (version 3.6.1) with default parameters. If autosomes had a positive Spearman's *r* value, the 100-kb bins with positive or negative PC1 were identified as compartment A or B, and otherwise were identified as compartment B or A, respectively. The A-B index was then calculated at 20-kb resolution using the publicly available code, get_ABindex.py (https://github.com/JiamanZhang/Lab_Porcine-Adiposes_paper_codes/tree/main/Lab_AB_compartment), as previously described[100]. A-B index represents the likelihood of a genomic segment interacting with the A or B compartments defined at 100-kb resolution, as described above. The 20-kb bins with positive or negative A-B indexes were considered A or B compartments, respectively.

The A/B compartment frequency of each 20-kb bin was identified as the proportion of that bin found in the A compartment across replicates/samples in each AT of a given group. The frequency of A/B compartment status can be roughly classified into three categories: high frequency (≥70%), low frequency (≤30%), and medium frequency (30–70%), according to their consistency across replicates/samples. We initially assessed changes in compartment frequency between treatment groups or ATs.

To compare the compartment status between groups/ATs in more detail, we defined A/B switches and A/B variables using the following pipeline: first, we defined a set of common A/B compartments (with >80% of individuals/replicates exhibiting the same chromatin status). The compartment status of a bin that exhibited opposite trends between groups/ATs was defined as an A/B compartment switch. We then identified regions with the same compartment status between groups/ATs, but with statistically significant differences in compartment scores (i.e., the A-B index) between groups/tissues (|ΔAB index| > 0.75 and *q* value < 0.05, Student's *t* test and adjusted FDR), which were then designated as A/B variables.

**TAD identification and analysis**. The deDoc[101] (version 1.0.0) program at default parameters was used to identify TADs in the 20-kb normalized contact matrices (generated by the Knight-Ruiz algorithm and quantile method) for each autosome. TADs shorter than 100-kb were removed. To characterize the strength of each TAD structure, we

calculated IS[102] (insulation score) using matrix2insulation.pl (version 1.0.0, https://github.com/dekkerlab/cworld-dekker) with the parameters (−v -is 260,000 -ids 200,000 -im mean -nt 0.1 -bmoe 0) and LBS[29] (local boundary score) using the get.samples.chr.LBS.value.py script (https://github.com/JiamanZhang/Lab_Porcine-Adiposes_paper_codes/tree/main/Lab_TAD_LBS/codes) with default parameters at 20-kb resolution.

We also measured the concordance of TADs across pairwise samples using the Jaccard index (a measure of the similarity between two sets of data), MoC overlap ratio[103] (a measure of the overlap between each pair of TADs that assesses the number of base pairs, and considers the overall size of both TADs), and VI scores[104] (a measure of the similarity of all subsets of the two TAD structures using a dynamic programming algorithm to compute the VI metrics).

**TAD boundary frequency**. To identify non-redundant boundaries to compare between ATs and between groups, we collapsed TAD boundaries with proximal genomic locations and reconstituted aggregated sets of non-redundant TAD boundary-enriched regions (200-kb regions bookended by TAD boundaries), as previously described with minor modifications[105]. We first calculated average insulation scores (IS) across each group for TAD boundaries, concatenated TAD boundaries from all replicates within a given comparison, and sorted them by there is in ascending order. Next, we picked one boundary from the top of the list and removed any remaining boundary within ±100 kb of the top boundary. The next boundary on the list and the process were then repeated until the entire list was complete.

Non-redundant boundaries were used as boundary centers to calculate/quantify frequencies based on the number of replicates in which they were observed in these boundary-enriched regions (±100 kb regions) for each treatment group (Supplementary Fig. 18a). TAD boundaries were also classified as high frequency (≥70%); low frequency (<30%); or medium frequency (30–70%).

**Features of variable or stable A/B compartments and TAD boundary regions**. We also characterized sequence elements embedded in regions of variable or stable A/B compartments or TAD boundaries, including CTCF motifs that were identified by FIMO[106] (version 5.1.1) with default parameters; phastCons and phyloP of pig were obtained by liftovering the corresponding values for the human genome version hg38 (downloaded from UCSC, http://www.genome.ucsc.edu) with pig reference genome Sscrofa11.1 using the UCSC LiftOver tool (version linux.x86_64); housekeeping genes and transposon elements (TE) downloaded from the previous study[6].

**Transposon element (TE) analysis**. Repeat Masker data of Sscrofa11.1 was downloaded from UCSC (https://hgdownload.soe.ucsc.edu/goldenPath/susScr11/bigZips/). For further analysis, we only retained four types of TEs (short interspersed nuclear elements [SINE], long interspersed nuclear element (LINE), long terminal repeat (LTR), DNA) [DNA transposons]. The sequence diversity of TEs was used to characterize the age of the boundaries.

**Ultra-deep Hi-C data pooling**. To further explore fine-scale chromatin structures at a high resolution, including CCCTC-binding factor [CTCF]-mediated loops and promoter-enhancer interactions (PEIs) for each AT across NC, WG, and WL groups, Hi-C reads obtained from replicates within NC and WL group were pooled. For the WG group, we defined sub-groups based on the similarity of their cell composition (including adipocytes, macrophages [M1 and M2 combined], and MVECs) across all 157 samples using K-means clustering. We empirically defined 2–3 sub-groups for a given AT in the WG group (sub-groups with 7 or more biological replicates were retained), and replicates within each WG sub-group were pooled. To ensure balance in the

pooling, the pooled data were down-sampled to about 2 billion read pairs, which generated contact matrices that reached a 1.5-kb resolution.

**Analysis of CTCF-mediated loops.** We used Fithic2 software[107,108] (version 2.0.7) with default parameters to identify loops at a 5 kb resolution, using *q* value < 0.01 as a cut-off. Loops with both anchors overlapping a CTCF motif and in the genomic range of 30-kb to 2-Mb were retained for further analysis. We next merged redundant paired-end loops within a neighboring genomic distance (distances between both anchor pairs less than a 5-kb bin) following protocols used in previous work, with minor modifications[10]. We first ranked all loops by their chromosome position and subsequently divided them into two groups based on whether they had even or odd ranks. We then used the pair-to-pair command in bedtools (version v2.25.0) to investigate overlaps in the boundaries between any paired loops from the two sets. Both anchors of loops in one set that overlapped or neighbored with anchors of loops in the other set were merged to form new loops with unified anchor regions. Loops having no overlap with any other loops were retained. The merged and retained loops were used as inputs for the next iteration. Iterations continued until the algorithm converged and no other paired-end loops could be merged.

**ChIP-seq library preparation and sequencing.** We performed ChIP-seq using antibodies against H3K27ac (a canonical histone mark of active enhancers) as previously described[109]. In brief, AT samples were fixed with 1% formaldehyde. The samples were then lysed, and chromatin was isolated on ice. Samples were sonicated to obtain soluble, sheared chromatin with an average DNA length of 200–500 bp. The 20 μl soluble chromatin was stored at −20 °C for sequencing as input DNA, while the 100 μl soluble chromatin was used for immunoprecipitation with the 5 μg H3K27ac (ab4729, Abcam) antibodies. For both input DNA and immunoprecipitated DNA, each ChIP-seq library was sequenced on the Illumina HiSeq X Ten platform to generate 150 bp paired-end reads.

**ChIP-seq data analysis.** H3K27ac ChIP-seq reads were mapped to the pig reference genome (Sscrofa 11.1) using BWA[110] (version 0.7.15). Next, PCR duplicates were removed using Samtools[111] (version 1.3.1). Peaks were called using the SICER[112] tool (version 0.1.1) with parameters (--windowSize 200 --gapSize 3 --mapq 0 --fragSize 250 --FDR 0.05). We performed the peak calling step for each sample bam file, and group bam files were pulled from all replicates. Peaks called from group bam files were retained if the peak was observed in at least 50% of all replicates and the overlap length was at least 50% of both pairwise peaks. Highly and moderately active enhancers were identified by the standard ROSE algorithm[113,114] (version 0.1) with parameters (-s 12500 -t 2000). Briefly, neighboring enhancer elements (within 12.5 kb) were defined by H3K27ac ChIP-seq peaks that were merged and ranked by the H3K27ac signal to identify an inflection point. Enhancers above the inflection point were considered highly active enhancer peaks, while those below the inflection point were considered moderately active enhancer peaks. Fold enrichment over control signal tracks was determined using the bdgcmp command in MACS2 (version 2.1.1.20160309) with default parameters.

**Identification and analysis of promoter-enhancer interactions (PEIs).** Normalized contact matrices (using Knight-Ruiz algorithm and quantile method) at a 5-kb resolution were analyzed using the PSYCHIC[34] algorithm (version 2018-01-05, https://github.com/dhkron/PSYCHIC) with default parameters to identify over-represented interactions in the promoter region. We retained high confidence PEIs with FDR values < 0.001 and interaction distance ≥40 kb. To explore the potential for transcriptional regulation by a predicted enhancer,

we calculated a regulatory potential score (RPS) for each gene based on the hypothesis that an enhancer's quantitative effect on a gene could depend on their spatial proximity. The RPS was calculated as:

$$\text{RPS} = \sum_{i=1}^{n} \log_{10}(l_n) \tag{2}$$

in which *n* represents the number of enhancers linked to a gene in the aforementioned high confidence PEIs and $l_n$ represents the normalized interactions (i.e., the observed contact frequency minus the expected contact frequency) for each PEI. The observed contact frequency of PEI was obtained from KR and quantile normalized contact matrices, and the expected contact frequency of PEI was obtained from expected contact matrices calculated by PSYCHIC[34]. To investigate PEI rewiring, we compared the RPS between ATs/groups; differential RPS genes were defined as those genes with FC [fold change] >1.5, |Δ| >2.

We also quantified the activity for putative enhancers (5-kb in length) involved in PEIs by status annotation identified by ROSE, which was then classified into three categories, including highly-active enhancers (covered by the H3K27ac peak), moderately-active enhancers (covered by the H3K27ac peak), and low-active enhancers (not covered by the H3K27ac peak). We used ROSE with a -t parameter set to 2000 to exclude promoter regions.

**Visualization of 3D chromatin structure.** The miniMDS (version 2018-09-27, https://github.com/seqcode/miniMDS) program was used to infer the 3D genome structures at the 5-kb normalized contact matrix (using Knight-Ruiz algorithm and quantile method)[115]. Pymol[116] (version 2.5.2) was used to visualize 3D coordinates.

**Evolutionary patterns of local spatial context across mammals**
**Local spatial context comparison across species.** Evolutionary patterns of local spatial context (i.e., IS) across multiple species (based on references genomes for pig [Sscrofa11.1], including humans [hg38], dogs [CanFam3.1], cats [felCat9], mice [mm39], rabbits [OryCun2.0], and sheep [Oar_v3.1]) were identified using the Phylo-HMGP model[43] (version 1, https://github.com/yangymargaret/Phylo-HMGP) with parameters (--num_states 30 -r 31). To compare expression levels between multiple species, TPM values were normalized using a previous published scaling method[117].

**Transcription factor (TF) binding motif analysis.** Genome-wide motifs were identified by FIMO[106] (version 5.1.1) with default parameters using the JASPAR 2016 core vertebrate motif database[118]. TF motif enrichment across different states was calculated using HOMER's findMotifs.pl (version v4.4, http://homer.ucsd.edu/homer/) separately, with a background of stable anchors[119].

**Comparative analysis between pigs and humans**
**Human data collection.** We prepared the in situ Hi-C, RNA-seq, and ChIP-seq libraries for the collected human adipose samples with the same procedures used for pig samples. We also downloaded publicly available Hi-C and RNA-seq datasets of three human SAT samples (see Supplementary Data 9 for details). We performed expression, A/B compartment, TAD boundary, and PEI analyses for four homologous ATs with the same analytical pipeline used to assess the pig data.

The collection and sequencing of human clinical samples complied with all relevant regulations regarding the use of human study participants approved by the Ethics Committee of Sichuan Provincial People's Hospital (No. 2018-212), and informed consent was obtained before the study. All study participants provided written informed consent in accordance with the Declaration of Helsinki.

**Inter-species comparison/mapping of PEIs.** We separately mapped the contacts of four ATs in pigs ("query" species) to the human genome

("reference" species) using the LiftOver model in the C-InterSecture tool (version 2020-09-14, https://github.com/NuriddinovMA/C-InterSecture) and vice versa. We only focused on comparable homologous promoters, with bins (5-kb) where the TSS location of genes could be lifted over using the UCSC LiftOver tool (version linux.x86_64) with the parameter "minMatch = 0.2" cross Sscrofa11.1 to hg38, and vice versa. Human and pig homolog files (release 102) were downloaded from BioMart (https://asia.ensembl.org/index.html).

**Inter-species enhancer sequence and usage conservation.** The conservation of enhancers that interacted with homologously comparable promoters (i.e., liftover) between human and pig genomes was assessed based on their sequence divergence and functional usage using C-InterSecture[120], according to the following criteria: (1) Enhancers that failed to liftover were recognized as sequence-specific. (2) Enhancers where liftover was successful and whose percentile scores (PS, calculated by normalized contact frequencies using C-intersecture[120]) exceeded 85 were defined as usage-conserved. (3) The remaining enhancers with successful liftover that had PS values ≤ 85 were defined as only sequence-conserved.

**Enrichment analysis of non-coding SNPs in enhancers.** Human SNPs were downloaded from the NHGRI-EBI GWAS catalog (https://www.ebi.ac.uk/gwas/). SNP records related to genome-wide haplotype association study (GWHAS) and SNP-by-SNP associations were discarded. Pig SNPs were lifted over from human SNPs using the UCSC LiftOver tool (version linux.x86_64). To investigate SNP enrichment around enhancers and non-coding regions that did not overlap with coding sequence regions of PCGs, we calculated enrichment scores[121] as follows:

$$\text{Enrichment score} = \frac{N_{\text{xsnp}}}{N_{\text{all}}} \times \frac{\text{BN}_{\text{all}}}{\text{BN}_x} \qquad (3)$$

in which, $N_{\text{xsnp}}$ represents the number of SNPs in X type regions (e.g., enhancer regions) and $N_{\text{all}}$ is the total number of SNPs on autosomes; $\text{BN}_x$ represents the base number of X type regions, and $\text{BN}_{\text{all}}$ is the base number of the sum of autosomes.

**Identification of expansion and contraction genes.** We used five species (humans, pigs, mice, rats, and cows) to identify expanded or contracted gene families using CAFE[122] (version 4.2.1, https://github.com/hahnlab/CAFE) with default parameters. A phylogenetic tree of five species was obtained from the TimeTree database (http://timetree.org). We obtained homologs (only PCGs) from five species (release 102) from BioMart using humans as a reference. Homologous gene networks among the five species were constructed by assessing the homology relationship between humans and other species. Genes with copy numbers (the number of gene IDs belonging to specific species in a family) that were greater (or lower) in humans than the average of the other four species and "Viterbi p values" less than 0.05 calculated by CAFE[122], were considered human expansion (or contraction) gene families.

**Functional enrichment analysis**
Gene Ontology (GO) and KEGG pathway functional enrichment analyses were performed using Metascape (version 3.5, http://metascape.org). Genes were mapped to their respective human orthologs, and the lists were submitted to Metascape for enrichment analysis based on the significant overrepresentation of GO biological processes (GO-BP) and KEGG-pathway categories. In all tests, all the annotated genes in the genome were used as the enrichment background. Only GO-BP or KEGG-pathway terms with resulting p values or FDR corrected p values <0.05 were considered significant and were depicted in the plots.

**Data collection for functional gene categories**
To further characterize the specialized functions of ATs in this study, we collected multiple a priori functional candidate PCGs and examined their expression patterns or chromatin status. PCGs involved in the "hypertrophy genes", "hyperplasia genes", and development-related *HOX* genes" and "homeobox family genes" were retrieved from public databases (Kyoto Encyclopedia of Genes and Genomes (KEGG) and Gene Ontology) and/or collected from the literature.

**Reporting summary**
Further information on research design is available in the Nature Portfolio Reporting Summary linked to this article.

## Data availability
The data that support this study are available from the corresponding authors upon reasonable request. The reference genome and gene annotation file (Sscrofa11.1, release 102) were downloaded from Ensembl (https://ftp.ensembl.org/pub/release-102/). A phylogenetic tree of five species was obtained from the TimeTree database (http://timetree.org). The raw and processed Hi-C, RNA-seq, and ChIP-seq data of pigs generated in this study are available at Gene Expression Omnibus (GEO) under the accession code "GSE206539". The raw Hi-C, RNA-seq, and ChIP-seq data of humans generated in this study are available at Genome Sequence Archive for Human (GSA-Human) under the accession code "HRA002514". The public Hi-C and RNA-seq data of nine adipose samples of pigs were downloaded from Sequence Read Archive (SRA) under the BioProject accession codes "PRJNA637678" and "PRJNA733023". The public Hi-C and RNA-seq data of three adipose samples of humans were downloaded from SRA under the BioProject accession code "PRJNA678123". The public Hi-C and RNA-seq data of the other five species for cross-species analysis were downloaded from SRA under the BioProject accession codes "PRJNA637678" and "PRJNA817154". Details are available in Supplementary Data 1, 6 and 9. Source data are provided with this paper.

## Code availability
All the code used for data analysis is available at https://github.com/JiamanZhang/Lab_Porcine-Adiposes_paper_codes and https://doi.org/10.5281/zenodo.7894973.

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

## Acknowledgements

We thank Dr. Isaac V. Greenhut for his valuable discussion and feedback on this manuscript. We thank the High-Performance Computing Platform of Sichuan Agricultural University, and Northwest A&F University for providing computing resources and support that contributed to this research. This work was supported by the National Key R&D Program of China (2020YFA0509500 to M.L., 2021YFA0805903 to L.J., 2022YFF1000100 to Q.T., and 2021YFD1300800 to L.L.), the Strategic Priority Research Program of CAS (XDA24020307 to Z.Z.), the Tackling Project for Agricultural Key Core Technologies of China (NK2022110602 to L.J.), the Sichuan Science and Technology Program (2021ZDZX0008, 2022NZZJ0028 and 2022JDJQ0054 to L.J. and Q.T., 2021YFYZ0009 and 2021YFYZ0030 to M.L., 2021YFS0008 and 2022YFQ0022 to L.L., 2022NSFSC1618 to J.L., 2022NSFSC0056 to K.L., and 2021YFH0033 to J.M.), the National Natural Science Foundation of China (U19A2036 and 32225046 to M.L., 32272837 to L.J., 32102507 and U22A20507 to J.L., 32102512 to K.L., and 32202630 to F.K.), the Special Investigation on Science and Technology Basic Resources of the MOST of China (2019FY100102 to Z.Z.), the China Agriculture Research System (CARS-35-01A to X.L.), the Beijing Natural Science Foundation (Z200021 to Z.Z.), the Major Science and Technology Projects of Tibet Autonomous Region (XZ202101ZD0005N to J.M.), the Opening Foundation of Key Laboratory of Pig Industry Sciences (22519C to F.K.), and the Ya'an Science and Technology Program (21SXHZ0022 to L.J.).

## Author contributions

M.L., L.J., and Z.Z. led the experiments and designed the analytical strategy. P.L., Y.W., L.J., C.L., Z.Ha., K.L., L.L., X.W., Z.Hu., and L.S. performed animal work and prepared biological samples. Y.Z., Xi.L., J.L., F.K., and J.M. constructed the sequencing library and performed sequencing, designed the bioinformatics analysis process. D.W., L.J., J.Z., D.L., J.B., X.F., and B.Z. performed the gene expression quantification analysis. D.W., J.Z., L.J., Y.L., Q.T., M.L., and Z.Z. performed Hi-C data analysis. J.B. and J.Z. performed 3D genome structures reconstruction. J.Z., J.B., and L.J. performed evolutionary analysis across species. D.W., J.Z., and L.J. performed inter-species comparative analysis of PEIs. L.J., D.W., J.Z., M.L., and Z.Z. wrote the paper. M.L., Z.Z., L.J., H.L., Yu.J., G.L., L.Z., Ya.J., G.T., B.F., L.G., and Xu.L. revised the paper.

## Competing interests

The authors declare no competing interests.

## Additional information

[1]Livestock and Poultry Multi-omics Key Laboratory of Ministry of Agriculture and Rural Affairs, College of Animal Science and Technology, Sichuan Agricultural University, Chengdu 611130, China. [2]Animal Breeding and Genetics Key Laboratory of Sichuan Province, Institute of Animal Genetics and Breeding, Sichuan Agricultural University, Chengdu 611130, China. [3]CAS Key Laboratory of Genome Sciences and Information, Beijing Institute of Genomics, Chinese Academy of Sciences and China National Center for Bioinformation, 100101 Beijing, China. [4]School of Life Science, University of Chinese Academy of Sciences, 100049 Beijing, China. [5]Sars-Fang Centre and MOE Key Laboratory of Marine Genetics and Breeding, College of Marine Life Sciences, Ocean University of China, Qingdao 266100, China. [6]Animal Molecular Design and Precise Breeding Key Laboratory of Guangdong Province, School of Life Science and Engineering, Foshan University, Foshan 528225, China. [7]School of Pharmacy, Chengdu University, Chengdu 610106, China. [8]Key Laboratory of Animal Genetics, Breeding and Reproduction of Shaanxi Province, College of Animal Science and Technology, Northwest A&F University, Yangling 712100, China. [9]Institute of Nephrology, Sichuan Provincial People's Hospital, University of Electronic Science and Technology of China, Chengdu 610072, China. [10]Institute of Animal Nutrition, Sichuan Agricultural University, Chengdu 611130, China. [11]Ya'an Digital Economy Operation Company, Ya'an 625014, China. [12]Pig Industry Sciences Key Laboratory of Ministry of Agriculture and Rural Affairs, Chongqing Academy of Animal Sciences, Chongqing 402460, China. [13]These authors contributed equally: Long Jin, Danyang Wang, Jiaman Zhang. ✉e-mail: zhangzhihua@big.ac.cn; mingzhou.li@sicau.edu.cn

