## [Peer Review File · Nature Communications]

Dynamic chromatin architecture of the porcine adipose tissues with weight gain and lossEditorial Note: Parts of this Peer Review File have been redacted as indicated to remove third-party material where no permission to publish could be obtained.

REVIEWER COMMENTS

Reviewer #1 (Remarks to the Author):

OVERVIEW

This manuscript presents an extensive analysis of gene expression and 3D chromatin conformation in porcine adipose tissues during a Weight-Gain/Weight-Loss experiment. This outstanding study features one of the largest Hi-C datasets ever generated, providing genome interaction matrices for 249 samples, along with the associated RNA-seq gene expression and H3K27ac ChIP-seq data. The authors realized several comparative analyses between Adipose Tissues (ATs) and between treatments groups (the latter with less success), characterized TAD and A-B compartment dynamics across samples, identified interesting Differentially Expressed Genes (DEG), Promoter-Enhancer Interactions (PEIs), and investigated evolutionary aspects of ATs with a comparative genomics study across mammals.

The manuscript is well written, with plenty of detailed and informative figures. Considering the relevance of the pig model for medical research, the study is of interest. The volume of data that has been generated and analyzed is remarkable. The study produced abundant and highly valuable results.

I only have a few concerns to mention, in particular one about potential confounding factors during the analysis, and another one about the way differences between treatment groups are presented.

MAIN POINTS

1) Possible confounding factors?

I am a bit concerned about 2 potential confounding factors in the comparative analyses.

- Sequencing depth.

According to Sup Data 1, the number of valid contacts in the Hi-C maps strongly differs between experimental groups, with an average of 342M, 291M, and 254M interactions for NC, WG, and WL respectively.

While this should not be an issue for compartment-related metrics (e.g. A-B index), it could however impact TAD-related and local compaction metrics (e.g. IS). TAD calling is known to be sensitive to matrix density. Therefore, I would like to be sure that the inter-sample normalization got rid of this potential issue.

In that regard, could the authors show the distribution of the following features within each replicate across groups? TAD size, IS score, A-B index, and CTCF loop size.

- Cellular composition.

Differences in cellular compositions between bulk tissue samples are known to affect comparative analyses. The authors investigated the issue and confirmed that ATs have a highly variable cellular composition: according to Fig S18a, adipocyte content can vary from ~26% to ~98% across samples. Such a high diversity might explain in part why the group factor does not seem to have a strong effect in the t-SNEs for example. It could also explain part of the results, including the immunity-related GO terms from the functional analyses. Note that Fig S18a shows a much higher proportion of macrophages in GOM vs. ULB for instance. Of course this is difficult to assess since the deconvolution analysis relies on the same expression data used for the functional analysis.

Could the authors estimate the relative proportion of adipocytes vs. immune cells in samples from the other groups? It would help to see the distribution of this ratio by AT and by group.

I am aware that WG is expected to induce an immune response, but I wonder to which extent some results from the comparative analyses might just be due to a difference in the cellular composition of

the compared samples instead of an actual modification of the gene expression program and/or chromatin rewiring inside the cells. Maybe the authors could check the expression patterns of the Fig 3h-i genes across sub-groups, in order to assess if their expression is widespread across sub-groups or more likely to be cell-type specific.

To deal with this, it might be possible to perform a multi-factorial Differential Gene Expression analysis to take into account the experimental variables during the comparison, including factors like tissue, group and cell type composition. I am not sure the same type of analysis could be performed to compare chromatin structures though. Another way to address this issue -if any- might be to perform single-cell experiments, but this might go beyond the scope of the project, depending on how this is addressed in the text.

The authors actually acknowledge and briefly mention this point in a sentence at the end of the discussion. Considering the goal of the study I believe that this is an important point that deserves more consideration throughout the manuscript.

2) Group effect vs. AT effect?

While major differences between ATs are clearly visible in terms of gene expression and chromatin structure, results do not show much impact of the treatment group (WGWL) globally, which is sometimes undermined in the text. No global effect is visible at the scale of the full transcriptome, TAD structure or chromatin loops. No clear signal in the t-SNEs, at least on the two first dimensions, or in the violin plots (Fig S4 & S5).

As a consequence, and considering my previous point about tissue heterogeneity, claiming for instance that "Rewiring of PEIs underpin transcriptional programs during AT response to body weight changes" (line 210-211 and 1298) sounds a bit of a stretch to me. Actually, according to Fig S7, the difference between samples from different groups does not seem much higher than between replicates. Several figures and numbers support this concern, including the relatively low number of genes (105) with covariation between RPS and gene expression. The nice examples shown by the authors in the article might be more the exception than the rule. To me this is not an issue about the data or the analysis, as differences between ATs look valid. It is more about how differences between groups are presented, in accordance with the supporting evidence. In my opinion, the contrast between the global effects of AT vs. group is actually an interesting outcome of the study.

Therefore, I believe that the following claims --and the ones alike-- are not strongly supported and should therefore be attenuated:

Line 10: "Dramatic transcriptomic and chromatin architectural changes were detected"

15: "high similarity" (=> "similarities"?)

71: "dramatic alterations during WGWL"

94: "revealed dramatic alterations in the WG and WL groups compared with the NC group."

100: "substantially diverged in the WG and WL groups compared to the NC group (Fig 1g-i), suggesting that..."

The second paragraph of the results in particular should be rephrased.

3) Methods

Methods would be better with more details. Software names, versions and parameters are often missing.

DETAILED REPORT (mostly minor points, with line numbers)

107: "ten genes related to VAT hypertrophy"

Before presenting the results of the GO functional analysis or focusing on specific subsets of genes, could the authors present the global results from the Differential Gene Expression analysis that is presented in Methods? How many DEG were found by edgeR between ATs and between groups? Could

they provide the list of DE Genes in the sup. data?

159: Compartment switching seems to be presented twice with different definitions (line 160 vs. 700). This is confusing.

167: "genes near compartments" => genes "in" compartments? How was the assignment made?

179: "the vast majority of which did not vary between replicates" => According to Fig S17b only a minority of them are in the High frequency category.

189: "genes near shifted TAD boundaries" => how is "near" defined?

189: "genes near shifted TAD boundaries showed more dramatic changes in expression than those near the common boundaries" => This is not clear to me in Fig 2h, as most of the comparisons do not show any significant change actually. The author should rephrase and maybe highlight the contrast with results from A/B compartments.

191: "These substantial differences" => same than above.

197: "EN1 expression in ULB is higher than in GOM ($P = 5.30 \times 10^{-5}$, Wilcoxon rank-sum test)"
Since a DGE analysis was performed using a proper normalization method and FDR correction, I do not understand the point of using a Wilcoxon test on TPM values without adjusting p-values. See also line 205 for instance.

219: "PEIs were preferentially located within TADs"
Given that PEIs are on average much smaller than TADs, how did the authors tested if the proportion of intra-TAD PEIs was significantly higher than expected by chance? Similarly, how was the enrichment with CTCF loops tested? I could not find the details in Sup note 4.

228: Unlike all previously mentioned features, PEI usage does not seem to show a higher similarity between replicates within ATs vs. groups (~36% consistent PEIs in both cases, as supported by Fig S19a). Fig 3c-d and Fig S19c suggest otherwise. Could the authors comment on that?

355: Fig S35 does not support that conclusion

382: "with rapidly evolving [...] than those" => a "more" might be missing

433: "supporting the 'ortholog conjecture'" => isn't it "questioning" more than supporting, as the orthologous conjecture seems to advocate for a higher conservation of orthologous vs. paralogous gene?

677: How was the quantile normalization performed? Tool name (if not Juicer), version, parameters?
Same question for the O/E transformation line 679.

692: "The A-B index was then calculated at a 20-kb resolution, as previously described"
Could the authors give more information about how the A-B index was computed exactly? Was this done using Juicer as mentioned above or with another tool?

707: Which matrices were used to call TADs? O/E matrices were used for the A/B compartment prediction but I guess that TAD finding was performed on raw or KR-normalized matrices. This is not precised here. How IS and LBS were computed is not precised either.
More generally, could the authors make sure to precise for each analysis: input data, software name, version, and parameters?

747: "were identified by FIMO102, phastCons, and phyloP (downloaded from UCSC)"

Could the authors be more specific about where to access phastCons and phyloP information for pig in UCSC?

894: How was CAFE used? Were the load/tree/lambda/report commands executed? Was a Newick phylogenetic tree provided, and if so, where was it obtained from?

905: When testing for functional enrichment of GO terms in the DE gene set, what was the background gene set? All the annotated genes or only the ones that were expressed in the ATs?

1256: I do not understand the Circos plots in Fig 2a. Also, I cannot read what is written in the inner circles, the font is too small.

Fig S4: How were HiCrep, Genomedito and QuASAR used exactly and on which data: raw interaction matrices, ICE and/or O-E normalized matrices?

Fig S25a: "Enhancers"

Reviewer #2 (Remarks to the Author):

This study has done very comprehensive investigation of potential regulatory mechanisms of obesity phenotypes in a pig model by primarily focusing on 3D genome chromatin interaction from anatomically distinct adipose tissues under different diet-induced condition. The results provide one of the largest high-resolution 3-D genome map in adipose tissues in farm animals and will be a great resource to identify potential genetic variants particularly in non-coding regions associated with obesity, an important human disease. The data generated could also provide many potential hypothesis that need be further investigated in the field.

Here are some specific comments:

1. There were significant phenotype and gene expression difference between weight gain and weight loss group compared to control group. If chromatin interaction plays a significant role in regulating gene transcription, some specific example to demonstrate how A/B compartment, TAD difference affect gene regulation would be desirable as the phenotypic difference due to diet is the most important biology this study is trying to investigate underlying regulatory mechanisms.
2. The statement in Line 100-101 "all substantially diverged in the WG and WL groups compared to the NC group" in Fig 1g-i is not accurate (at least from figures presented, a different PCA plot only using NC, WG and WL maybe needed for this purpose).
3. Line 144-146, meaning is not clear.
4. Line 157 "most A/B compartments were invariable within replicates" does this mean among individuals pigs within each treated group and within each tissue?
5. For Fig. 2b, it seems RPS score could be high at 20-30, why the range of RPS is only up to 5?
6. Fig. 2e, it seems there is no positive correlation between TPM and RPS as presented in Fig 2b between RAD and ULB (totally negative correlated). In addition, the interaction between promoter and activities of enhancers interacting with promoters seemed not reflect the RPS score in the figure. Again, is there any example to compare between different treatment groups as they are more biologically important.
7. For Fig. 3i, for most genes, RPS is positively correlated with TPM, but for some genes, totally opposite, any explanation on this (C5AR1, CXCL8, HCK etc.)? it would be great to provide a figure for some of genes with TPM from low to high or high to low in the order of WG, NC, WL to demonstrate how A/B and TAD information in respective tissue, treatment groups, their conservation information across multiple mammals in potential promoters and its interacting enhancers contribute to gene regulation.

Reviewer #3 (Remarks to the Author):

Authors used a pig model to analyze the multi-scale structural dynamics of the 3D genome. They found the dramatic transcriptomic and chromatin architectural changes among the four ATs under different nutritional conditions. Then they revealed high similarity in the regulatory circuitry of genes responsible for the obesity phenotype and identified non-conserved elements in species-specific gene sets that underpin AT specialization. They also provide a tool for discovering obesity-related regulatory elements. Summary, this research provides transcriptomic and chromatin architecture of pig as a model system and compares it to other species. They confirmed the phenotype and underlying mechanisms of obesity development in humans and rodents. This study is interesting and well designed, however there are still some problems, which need to be solved before it is considered for publication.

1. The weight loss pig was getting from the weight gain pig rather than from the normal pig. How about the obesogenic and metabolic memory mechanism of getting weight loss from normal pigs?
2. In this study, some Hi-C and transcriptome data were downloaded from the public platform. So whether they come from the same individual? If not, there will be large bias in the integration analysis due to experimental conditions and individual differences.
3. Whether there are batch effects in Hi-C data is not discussed. If so, how to conduct batch removal? The paper needs to explore the batch effect of Hi-C data sampled from different batches.
4. 6. Section "Gene duplication accompanied by rapidly evolving enhancers" mentions that there are many of specific expanding genes families in human, which are correspondingly contracted in porcine genomes. The effects of the contractions on pigs should be discussed
5. In fig.1g-i, t-SNE clustering of different treatment groups is not clearly distinguished.
6. In section "Rewiring of PEIs underpins the AT transcriptional program response to different nutritional conditions", the detail definition and quantitative characteristics of "Rewiring of PEIs" should be added.
7. Is promoter bias excluded when using ROSE to find enhancers? It is recommended to set the -t parameter to remove the promoter region.
8. In line 825, I wonder what the O-E stands for here. It's seemed that the paper does not explain this abbreviation.
9. In line 824, how to define the number of enhancers linked to a gene ?

Reviewer #4 (Remarks to the Author):

The authors investigated transcriptome and chromatin architecture in two types of adipose tissues (ATs) - subcutaneous and visceral (from 3 different depots), using an adult miniature pig as model organism for human obesity. Novelty of this study is limited, since number of papers have been published about transcriptome of porcine ATs as well as chromatin organization in ATs. It is quite strange that Authors have not cited previous papers on this topic (e.g. DOI: 10.1186/s40104-022-00679-2). A new aspect of the study is comparison of chromatin architecture in adipose tissues from animals on different diets – normal diet (NC) as a control, a high-fat diet (WG) and calorie-restricted diet (WL). However, design of nutritional experiment has many failures. The WL group was created from WG group and the group of animals was subjected to nutritional regimen for next 12 weeks. The main weakness of such study design is different age of animals/time of keeping animals, which can significantly affect genome functioning. There is also no explanation why the 3 groups differ in size. Application of two-tailed t-test for comparison (Fig.1), when 3 groups are object of interest is not justified. Differences in phenotype (body weight, adipocytes size et) are visible only when compare WL with WG, no NC. All other analysis (transcriptome and chromatin studies) are referenced to the NC. Since, most of the presented results concerns different nutritional conditions, improperly conducted nutritional experiment, may affect obtained data. There are many other comments to the manuscript, e.g. the title is not appropriate - what the word "dynamics" is supposed to mean - it would have to

refer to the variability over time, which had no place in this work; the abstract is not very uninformative regarding the results obtained; in the introduction part no previous work done on pigs has been cited, etc. Concluding, due to methodological weakness of the study, the manuscript cannot be recommended for publication in present form.

Detailed responses to reviewers

All comments provided by reviewers are given in gray italics, and our responses are in black. All revisions in the manuscript are marked in red.

Reviewer #1

Comment 1-1:

This manuscript presents an extensive analysis of gene expression and 3D chromatin conformation in porcine adipose tissues during a Weight-Gain/Weight-Loss experiment. This outstanding study features one of the largest Hi-C datasets ever generated, providing genome interaction matrices for 249 samples, along with the associated RNA-seq gene expression and H3K27ac ChIP-seq data. The authors realized several comparative analyses between Adipose Tissues (ATs) and between treatments groups (the latter with less success), characterized TAD and A-B compartment dynamics across samples, identified interesting Differentially Expressed Genes (DEG), Promoter-Enhancer Interactions (PEIs), and investigated evolutionary aspects of ATs with a comparative genomics study across mammals.

The manuscript is well written, with plenty of detailed and informative figures. Considering the relevance of the pig model for medical research, the study is of interest. The volume of data that has been generated and analyzed is remarkable. The study produced abundant and highly valuable results.

I only have a few concerns to mention, in particular one about potential confounding factors during the analysis, and another one about the way differences between treatment groups are presented.

Response 1-1:

We very much appreciate the reviewer's careful consideration of our study. We strive to provide the most clear and coherent description of our findings. The authors are also grateful for the reviewer's constructive critique which has helped guide some improvements to our work.

Comment 1-2

1) Possible confounding factors?

I am a bit concerned about 2 potential confounding factors in the comparative analyses.

- Sequencing depth.

According to Sup Data 1, the number of valid contacts in the Hi-C maps strongly differs between experimental groups, with an average of 342M, 291M, and 254M interactions for NC, WG, and WL respectively.

While this should not be an issue for compartment-related metrics (e.g. A-B index), it could however impact TAD-related and local compaction metrics (e.g. IS). TAD calling is known to be sensitive to matrix density. Therefore, I would like to be sure that the inter-sample normalization got rid of this potential issue.

In that regard, could the authors show the distribution of the following features within each replicate across groups? TAD size, IS score, A-B index, and CTCF loop size.

Response 1-2

Thanks for this valuable suggestion. We agree with the reviewer's concerns about the possible influence of sequencing depth on local compaction metrics. In the original manuscript, we normalized the raw contact matrix to minimize intra- and inter-sample biases using the Knight-Ruiz and quantile normalization methods, respectively (see **Methods section 'Initial processing of Hi-C data' for details**).

For TAD identification, we used deDoc software, which showed little sensitivity to sequencing depth or matrix density ¹. In brief, deDoc utilizes structural information theory for TAD identification. deDoc treats the Hi-C contact matrix as a representation of a graph, partitioning the graph into segments with minimal structural entropy. As shown in **Figure R1**, the numbers and lengths of TADs predicted with only sparse input data (for example, only 10% of the full dataset) are consistent with the original predictions generated with the full

dataset, thus supporting the robustness TAD identifications by deDoc despite differences in coverage among groups in the input data.

Figure R1. deDoc performance in TAD identification at different coverage levels of input data. Different proportions (50, 25, 10, 1 and 0.1%) of data from a publicly available dataset (Rao et al.², chromosome 22 in GM12878 cells) were sampled. We compared the number of domains **(a)**, the lengths of the domains **(b)**, and weighted similarities **(c)** of TADs identified in the whole and down-sampled dataset. The error bars indicate S.D. from 50 replications of each experiment. **(adapted from results reported in our previous work¹)**

For PEI and CTCF loop identification, we pooled data from replicates and down-sampled to approximately 2 billion read pairs for all pairwise comparisons, which minimized the effect of variability in data coverage.

According to the reviewer's suggestion, we have added the distribution of A-B index, TAD size, IS score, and CTCF loop size for replicates across groups to the revised manuscript (**New Supplementary Fig. 2**). We found that the distributions showed high similarity across groups, suggesting that the original sequencing depth had relatively little effect and comparisons were unbiased across groups, further supporting the reliability of our results. We again thank the reviewer for helping us to test the rigor of our analysis and quality controls.

New Supplementary Fig. 2. The distribution of A-B index (a), IS (b), TAD size (kb) (c), and CTCF loop size (kb) (d) proportions across groups for a given tissue. Colored lines represent mean value across replicates and shading around the mean represents standard deviation across replicates.

Comment 1-3

- Cellular composition.

Differences in cellular compositions between bulk tissue samples are known to affect comparative analyses. The authors investigated the issue and confirmed that ATs have a highly variable cellular composition: according to Fig S18a, adipocyte content can vary from ~26% to ~98% across samples. Such a high diversity might explain in part why the group factor does not seem to have a strong effect in the t-SNEs for example. It could also explain part of the results, including the immunity-related GO terms from the functional analyses. Note that Fig S18a shows a much higher proportion of macrophages in GOM vs. ULB for instance. Of course this is difficult to assess since the deconvolution analysis relies on the same expression data used for the functional analysis.

Could the authors estimate the relative proportion of adipocytes vs. immune cells in samples from the other groups? It would help to see the distribution of this ratio by AT and by group.

I am aware that WG is expected to induce an immune response, but I wonder to which extent some results from the comparative analyses might just be due

to a difference in the cellular composition of the compared samples instead of an actual modification of the gene expression program and/or chromatin rewiring inside the cells. Maybe the authors could check the expression patterns of the Fig 3h-i genes across sub-groups, in order to assess if their expression is widespread across sub-groups or more likely to be cell-type specific.

To deal with this, it might be possible to perform a multi-factorial Differential Gene Expression analysis to take into account the experimental variables during the comparison, including factors like tissue, group and cell type composition. I am not sure the same type of analysis could be performed to compare chromatin structures though. Another way to address this issue -if any- might be to perform single-cell experiments, but this might go beyond the scope of the project, depending on how this is addressed in the text.

The authors actually acknowledge and briefly mention this point in a sentence at the end of the discussion. Considering the goal of the study I believe that this is an important point that deserves more consideration throughout the manuscript.

Response 1-3:

We thank the reviewer for these very thoughtful comments. They have indeed raised a very important concern about a possible confounding effect inherent to comparative analyses of data obtained from bulk tissue samples with varying cellular compositions.

First, it warrants mention that adipose tissues are known to undergo substantial changes in cellular composition during the processes of weight gain or loss³. Nonetheless, bulk Hi-C data only provide the average features of chromatin architecture at the cell population scale. Single-cell resolution is necessary to determine the extent to which cellular heterogeneity in adipose tissue contributes to the observed differential signals in chromatin features.

Being aware of these issues, we developed deCOOC, a convolutional neural network-based cellular deconvolution model, to specifically address the complex task of mapping Hi-C in tissues according to cellular composition (manuscript submitted to *Nucleic Acids Research*). This model enables

deciphering of cell type-specific genome architecture and dynamics based on bulk 3D genomic data from heterogeneous tissues. For example, by applying deCOOC to bulk Hi-C data from visceral (i.e., GOM) and subcutaneous (i.e., ULB) adipose tissues, we found that the characteristic chromatin features of macrophages in these two anatomical sites were distinctly tied to different physiological functions. That is, active chromatin regions in macrophages in GOM were more significantly enriched with inflammation-related GO terms than those in ULB. These results were consistent with the well-documented inflammation-related activity of macrophages in visceral adipose tissue⁴.

In this current study, we also tried to determine the extent to which the observed differences between treatment groups or ATs were due to differences in cell composition rather than transcriptomic regulatory changes within cells.

As the reviewer mentioned, the immunity-related GO terms from the functional analyses could indeed possibly reflect the higher proportion of immune cells in each AT in comparisons between the NC and WG groups. To address this potential issue, we used the transcriptomic data to deconvolute the relative proportion of three cell types (including adipocytes, macrophages [M1/M2 combined], CD4, and microvascular endothelial cells [MVEC]) for the NC, WL and WG groups. As expected, we found a relatively higher proportion of macrophages in GOM than in ULB, as well as a relatively greater proportion of macrophages in the WG group compared to the NC group for each AT (except GOM) (**new Supplementary Fig. 41a**).

However, the same trends could also be seen even after controlling for cell composition. After clustering samples based on their transcriptome-deconvoluted cell composition, we found that one group of samples (subgroup 5 in WG group, **Supplementary Fig. 19a**) had a nearly identical cell composition under all conditions (**new Supplementary Fig. 41b**). By limiting our analysis to that sample group, we could thus exclude the potential influence of variability in cell composition. First, we found that the differences in transcriptomic profiles and compartmentalization were markedly greater between ATs than between treatment groups in t-SNE analysis (**new Supplementary Fig. 41c,d**).

New Supplementary Fig. 41. Comparison of differences in gene expression and A/B compartmentalization between adipose depots and between treatment groups with similar cellular composition.

a The cellular proportions of adipocytes, microvascular endothelial cells (MVECs), and macrophages during weight gain or weight loss for each adipose depot. Dots represent mean values across replicates. The interval lines show standard deviation across replicates.

b All 64 samples with highly similar cell composition (including adipocytes, macrophages, and MVECs) of four ATs in the WG group and ULB from all three treatment groups. The above numbers indicate the proportion of each cell type in each AT across groups. Data show means \pm SD.

c t-SNE clustering of the 64 samples with similar cell compositions using gene expression data (left). In the t-SNE plot, ellipses indicate AT samples with similar profiles, constructed at a probability of 0.85. Violin plot shows the Euclidean distance of gene expression between samples (middle). The percentage distribution of projection distance (derived from t-SNE plot) between each dot and a given line ($y=kx$, $k=-3$) across groups in ULB (right).

d t-SNE clustering of the same 64 samples using A-B index (left). Ellipses indicate AT samples with similar profiles, constructed at a probability of 0.85. Violin plot shows the Euclidean distance of A-B index between samples (middle). The percentage distribution of projection distance (derived from t-SNE plot) between each dot and a given line ($y=kx$, $k=300$) across groups in ULB (right).

In addition, we also performed differential expression (DE), A/B compartment, and RPS analyses between groups and between ATs. Functional enrichment analysis of differentially expressed genes (DEGs) revealed that inflammation-related genes were differentially up-regulated in each AT type of the WG group compared to the NC group (**new Supplementary Fig. 42a**). This enrichment for inflammation-related DEGs was also evident in A/B compartmentalization and RPS analysis (**new Supplementary Fig. 42c**). These results were strikingly similar to our original findings (**Supplementary Fig. 3, 14 and 20e**), suggesting that HFD-induced obesity is associated with an inflammatory response transcriptional program and/or chromatin rewiring in adipose cells independent of the pro-inflammatory effects related to increased populations of inflammatory or immune cells in adipose tissues that occur after weight gain. These results were also consistent with a recent single-nucleus resolution study in mice that showed HFD-induced obesity increases the inflammatory profile of mesothelial and endothelial cells in epididymal adipose tissue³, and contributes to a general tissue-level inflammatory state in obese mice.

Additionally, our comparative analysis between ATs recapitulated the well-characterized differences between VATs and SATs, particularly in their respective metabolically harmful or protective roles (**new Supplementary Fig. 42d,e**).

As recommended by the reviewer, we have added some further interpretation of our results to the Discussion section addressing this issue.

‘By limiting our comparative transcriptomic, A/B compartment, and RPS analyses to a subset of samples with relatively similar cell composition, we could exclude the influence of variation in cell composition (e.g., greater inflammation-related gene expression due to larger macrophage populations). We observed that inflammation-related gene activity increased in adipose tissues after weight gain, suggesting that HFD-induced obesity could indeed

lead to an elevated inflammatory transcriptional response and/or chromatin rewiring in adipose cells.’ (Discussion: page 18, lines 554-562)

New Supplementary Fig. 42. Functional enrichment for differentially expressed genes (a,b), A/B compartmentalization (c,d), and RPS changes (e) between groups or between ATs with similar cell composition.

a Plot showing the top 20 enriched GO terms of DE PCGs in NC-WG and WG-WL comparisons for ULB. The names highlighted in red for each pairwise comparison indicate the group in which the genes were highly expressed. GO term color scheme: dark green, metabolism-related terms; pale green, extracellular matrix (ECM) organization-related

terms; orange, inflammation- and immunity-related terms; and others ($n = 38$, not shown in the plot). Dot size is proportional to the number of enriched genes; dot color represents the $-\log_{10}(P\text{-value})$ (unadjusted). P values were calculated based on a one-sided accumulative hypergeometric test.

b Plot showing the top 10 enriched GO terms of DE PCGs in pairwise comparisons of AT with similar cellular composition in the WG group. Pairwise comparisons are between VAT and SAT or within VATs. Names highlighted in red indicate the tissue in which the genes were highly expressed. GO term color scheme: dark green, metabolism-related terms; pale green, extracellular matrix (ECM) organization-related terms; orange, inflammation- and immunity-related terms; and others ($n = 86$, not shown in the plot). Dot size is proportional to the number of enriched genes; dot color represents the $-\log_{10}(P\text{-value})$ (unadjusted). P values were calculated based on a one-sided accumulative hypergeometric test.

c,d Plot showing the top 20 enriched GO terms in comparisons between groups (**c**) and between ATs (**d**) with similar cellular composition. The presentation is the same as in **a** and **b**, but show A/B compartments of genes. Other terms (27 terms for **c**, 87 for **d**) are not shown.

e, Plot showing the top 20 enriched GO terms in comparisons between ATs with similar cellular composition. The presentation is the same as that in **a** and **b**, but for AT-specific genes with covariation between RPS and gene expression. Other terms ($n = 63$) are not shown. Number of group-specific genes with covariation between RPS and gene expression were generally fewer than 50, and thus insufficient for functional enrichment analysis.

Comment 1-4

2) Group effect vs. AT effect? While major differences between ATs are clearly visible in terms of gene expression and chromatin structure, results do not show much impact of the treatment group (WGWL) globally, which is sometimes undermined in the text. No global effect is visible at the scale of the full transcriptome, TAD structure or chromatin loops. No clear signal in the t-SNEs, at least on the two first dimensions, or in the violin plots (Fig S4 & S5).

As a consequence, and considering my previous point about tissue heterogeneity, claiming for instance that "Rewiring of PEIs underpin transcriptional programs during AT response to body weight changes" (line 210-211 and 1298) sounds a bit of a stretch to me. Actually, according to Fig

S7, the difference between samples from different groups does not seem much higher than between replicates. Several figures and numbers support this concern, including the relatively low number of genes (105) with covariation between RPS and gene expression. The nice examples shown by the authors in the article might be more the exception than the rule. To me this is not an issue about the data or the analysis, as differences between ATs look valid. It is more about how differences between groups are presented, in accordance with the supporting evidence. In my opinion, the contrast between the global effects of AT vs. group is actually an interesting outcome of the study.

Response 1-4

We thank the reviewer for these thoughtful comments. We completely agree that the objectively greater difference between ATs than between groups is, in fact, a reasonable and interesting outcome. In our experiment, pigs fed with HFD became markedly obese, with an overall weight nearly twice that of animals fed the normal diet after 22 weeks, while obese pigs treated with 12 weeks of caloric restriction showed a dramatic reduction in body weight. These obvious, dramatic changes in weight and adipose tissue phenotype/morphology (such as adipocyte size) suggested that the experiment was successful and led us to explore the underlying functional changes.

However, as mentioned by reviewer, we only found 27.13 Mb (~1.20% of the genome) regions of a given AT exhibited changes in A/B compartmental status or showed significant differences in A-B index between nutritional treatments. These changes were remarkably less dramatic than the changes in transcriptional program or 3D genome organization associated with malignant cancer (e.g., ~20% of the total genome switches A/B compartments in myeloma compared to corresponding normal tissue)⁵ or cell differentiation (e.g., ~20% of the total genome undergoes A/B compartment rearrangement during differentiation from embryonic cells into fibroblasts in humans)⁶. Similarly, relatively few changes in TAD boundaries (average 0.18%, ~22 boundaries) were detected in each AT across treatment groups. Therefore, although unexpected, it appears reasonable that AT undergoes fewer changes through

the relatively mild process of nutritional intervention than that observed in the more dramatic processes of cancer or cell differentiation in early development.

As suggested, we have revised our presentation of the differences between treatment groups to better illustrate this point (see **revised Fig.1**).

Revised Fig. 1. Divergence in transcriptomic and chromatin architecture dynamics among distinct ATs associated with changes in body weight.

a Schematic overview of the experimental design for dietary treatments.

b Distribution of body weight of pigs ($n = 56$) during progressive weight gain for all 22 weeks, observed every 2 weeks (12 time points, shown in different colors). Dotted lines indicate the average and ratio of weight at the first and last time point relative to the weight at baseline (0 weeks).

c Distribution of body weight of pigs ($n = 10$) during progressive weight loss for 12 weeks

(from 23rd to 34th week), observed every 2 weeks (6 time points, shown in different colors). Dotted lines indicated the average and ratio of weight at the first and last time point relative to the weight at the 22nd week.

d Histogram of body weight at slaughter for NC, WG, and WL groups. WG: weight gain, WL: weight loss, NC: normal diet. Data are presented as means \pm SD (NC, $n = 12$; WG, $n= 46$; WL, $n=10$). P values were determined by Wilcoxon rank-sum test.

e Sources of adipose tissues: one SAT (ULB: upper layer of backfat) and three VATs (GOM: greater omentum, MAD: mesenteric adipose, RAD: retroperitoneal adipose). SAT: subcutaneous adipose tissue, VAT: visceral adipose tissue.

f Histogram of adipocyte volumes during weight gain or loss for each adipose depot (top). Spheres show relative adipocyte volume, with the scale shown on the right. (middle) Fold changes in adipocyte volume in WG relative to NC (left) and in WL relative to WG (right) are shown below (bottom). Data are presented as mean values \pm SD. Statistical significance was determined using a Wilcoxon rank-sum test.

g–i Comparison of variation in gene transcription (**g**), AB compartment (**h**), and IS (**i**) between adipose depots and between groups. t-distributed stochastic neighbor embedding (t-SNE) clustering of samples. In t-SNE plots, ellipses indicate AT samples with similar profiles, constructed at a probability of 0.85.

j–l The proportional distribution of projection distance for t-SNE plots of gene expression in **g** (**j**), A-B index in **h** (**k**), and IS in **i** (**l**) between each dot and a given line ($y = kx$, $k = -0.3$, -1 , -5 for gene expression, A-B index and IS, respectively) across groups.

We have also revised our statement asserting that ‘Rewiring of PEIs underpin transcriptional programs during AT response to body weight changes’ to ‘**Rewiring of PEIs and associated transcriptional changes and response to body weight changes in different ATs.**’. (main text: page 8, lines 218-219)

As shown in **Supplementary Fig. 8**, the differences between samples from different groups does not appear much greater than that between replicates. Therefore, we have included additional statistical analysis in the **revised Supplementary Fig. 8**, and modified the corresponding description in the main text (main text: page 6, lines 148-152).

‘Using the distance between samples in the t-SNE as a metric, we found that the average distance between replicates was slightly less than that between treatment groups (Wilcoxon rank-sum test, $P < 1.96 \times 10^{-9}$), and markedly less

than that between pairwise ATs (Wilcoxon rank-sum test, $P < 2.2 \times 10^{-16}$) (Supplementary Fig. 8), suggesting considerable intra-group heterogeneity.'

Revised Supplementary Fig. 8. Correlation of gene expression, AB compartmentalization, and local spatial context (IS value) for pairs of samples between replicates, between tissues, and between groups. Statistical significance (P values) was determined using Wilcoxon rank-sum test. A-B index: between replicates vs. between groups, $P < 2.2 \times 10^{-16}$; between replicates vs. between ATs, $P < 2.2 \times 10^{-16}$; IS value: between replicates vs. between groups, $P < 2.2 \times 10^{-16}$; between replicates vs. between ATs, $P < 2.2 \times 10^{-16}$; Expression: between replicates vs. between groups, $P = 1.96 \times 10^{-9}$; between replicates vs. between ATs, $P < 2.2 \times 10^{-16}$.

Comment 1-5

Therefore, I believe that the following claims --and the ones alike-- are not strongly supported and should therefore be attenuated:

Response 1-5

Thanks for these astute suggestions. We have made the corresponding changes point by point below.

Comment 1-6

Line 10: "Dramatic transcriptomic and chromatin architectural changes were detected"

Response 1-6

We have reorganized this sentence and rewritten the full abstract (**main text: page 1, lines 4-23**).

‘To identify the regulatory mechanisms of three-dimensional (3D) genome architecture underlying obesity phenotypes in anatomically distinct adipose tissues (ATs), we used an adult **female** miniature pig model with diet-induced weight gain/weight loss to generate 249 high-resolution *in situ* Hi-C chromatin contact maps of subcutaneous AT and three visceral ATs. **Investigation of transcriptomic and chromatin architectural changes among the four ATs under different nutritional treatments showed multi-level remodeling of chromatin architecture that underpins transcriptomic divergence in ATs. These changes are potentially linked to progressive metabolic risks in obesity development (e.g., increasing inflammation) and the existence of persistent obesogenic memory even after caloric restriction-induced weight loss.** Analysis of chromatin architecture among subcutaneous ATs of different mammals suggested the presence of transcriptional regulatory divergence that could be responsible for phenotypic, physiological, and functional differences in ATs. Analysis of regulatory element (enhancer) conservation in all four ATs in pigs and humans revealed **similarities** in the regulatory circuitry of genes responsible for the obesity phenotype and identified non-conserved elements in species-specific gene sets that underpin AT specialization. This work provides an integrated, data-rich tool for discovering obesity-related regulatory elements through comparison of the 3D genome architecture of humans and pigs.’

Comment 1-7

15: "high similarity" (=> "similarities"?)

Response 1-7

Changed as suggested.

Comment 1-8

71: "dramatic alterations during WGWL"

Response 1-8

We have deleted 'dramatic'.

Comment 1-9

94: "revealed dramatic alterations in the WG and WL groups compared with the NC group."

Response 1-9

We have deleted 'dramatic'.

Comment 1-10

100: "substantially diverged in the WG and WL groups compared to the NC group (Fig 1g-i), suggesting that..."

Response 1-10

We have deleted 'substantially'.

Comment 1-11

The second paragraph of the results in particular should be rephrased.

Response 1-11

We have made corresponding revisions and carefully checked all analogous issues across the second paragraph.

Comment 1-12

3) Methods

Methods would be better with more details. Software names, versions and parameters are often missing.

Response 1-12

We have carefully checked for such issues throughout the entire methods section and added more detail, including versions and parameters for each software used in this study.

Comment 1-13

DETAILED REPORT (mostly minor points, with line numbers)

107: "ten genes related to VAT hypertrophy"

Before presenting the results of the GO functional analysis or focusing on specific subsets of genes, could the authors present the global results from the Differential Gene Expression analysis that is presented in Methods? How many DEG were found by edgeR between ATs and between groups? Could they provide the list of DE Genes in the sup. data?

Response 1-13

Thanks for this suggestion. We have added the global results of transcriptome analysis and the number of DEGs in **Supplementary Note 1**.

We also added **Supplementary Data 2**, which contains a detailed list of all DEGs identified by edgeR in comparisons between ATs and between groups.

*To characterize the transcriptomic differences among AT responses to changes of body weight, we sequenced 245 paired-end rRNA-depleted RNA-seq libraries from the AT samples (~11.84 gigabases [Gb] of high-quality sequences per library; ~3.30 terabases [Tb] total) (**Supplementary Data 1**). The transcriptomic variations of protein-coding genes and other two transcripts with essential regulatory roles (*i.e.*, long noncoding RNAs and transcripts of uncertain coding potential) (**Supplementary Fig. 6**) highlight that the differences between ATs (especially between SAT and VATs) extend beyond weight changes, and indicate that changes in chromatin architecture are correlated with shifts in transcriptional activity. Analysis of differentially expressed genes (DEGs) identified an average of ~1409 DEGs between pairwise ATs, especially between SAT and VATs (~1728 DE genes). An average of ~1322 DEGs were identified between the WG and NC groups, while ~1379*

DEGs were identified between the WG and WL groups for each AT (Supplementary Data 2).

We then performed functional enrichment analysis for differentially expressed protein-coding genes. For a given AT, genes (~ 643) that were up-regulated in WG compared to NC were primarily involved in immune and inflammation.....genes (~706) that are down-regulated in ATs of WL compared to WG were mainly involved in metabolism.....Under three nutritional conditions, genes that specifically were up-regulated in VATs (~804) and SAT (~920) were separately involved in immune and inflammatory processes (such as 'regulation of T cell cytokine production' and 'chemotaxis') and lipid metabolism processes.....' (Supplementary Note 1: page 6, lines 132-158)

Comment 1-14

159: Compartment switching seems to be presented twice with different definitions (line 160 vs. 700). This is confusing.

Response 1-14

Thanks for pointing out this point of confusion. Changes in compartmental status across ATs and among treatment groups were evaluated using two criteria, including compartment frequency (as described in the **Main text: page 6, lines 161-168**) and A/B switches combined with A/B variables (described in **Supplementary Note 2**). To improve clarity, we have replaced the term 'switching' with 'changes' and added a corresponding description in **Supplementary Note 2**. We also rephrased the content of '**A/B compartment determination and analysis**' in the Methods section.

'Moreover, compartmental status remained generally unchanged across ATs and treatments/groups, with **changes** (i.e., frequency of active compartment A status changed from high to low, or reversely) observed in an average of 1.31% (1489 bins of 20-kb length) and 0.29% (327) of bins, respectively (**Fig. 2a, b and Supplementary Data 3**).' (main text: page 6, lines 164-168)

'In addition to the analysis of compartment frequency changes between ATs or groups, we used a more detailed method to quantify these changes. We defined

the sets of SAT-restricted highly accessible regions (179.48 Mb; or ~7.92% of the genome) under different nutritional conditions that specifically have A compartment status (*i.e.*, A/B switches) or significantly higher A-B index against its counterpart in VATs (Δ A-B index > 0.75, Student's *t*-test, FDR < 0.01) (Supplementary Fig. 9–11).’ (Supplementary Note 2: page 7, lines 163-169)

‘A/B Compartment determination and analysis

A/B compartments at 20-kb resolution were identified using both principal component analysis (PCA) and A-B index, as previously described⁹⁸. PCA was performed to generate PC1 vectors for each autosome per sample at 100-kb resolution using the ‘prcomp’ function in R (version 3.6.1) with default parameters. Spearman’s coefficient *r* was calculated between PC1 and genomic characteristics, including gene density and GC content, for each autosome using the ‘cor’ function in R (version 3.6.1) with default parameters. If autosomes had a positive Spearman’s *r* value, the 100-kb bins with positive or negative PC1 were identified as compartment A or B, and otherwise were identified as compartment B or A, respectively. The A-B index was then calculated at 20-kb resolution using the publicly available code, `get_ABindex.py` (https://github.com/JiamanZhang/Lab_Porcine-Adiposes_paper_codes/tree/main/Lab_AB_compartment), as previously described⁹⁸. A-B index represents the likelihood of a genomic segment interacting with the A or B compartments defined at 100-kb resolution, as described above. The 20-kb bins with positive or negative A-B indexes were considered A or B compartments, respectively.

The A/B compartment frequency of each 20-kb bin was identified as the proportion of that bin found in the A compartment across replicates/samples in each AT of a given group. The frequency of A/B compartment status can be roughly classified into three categories: high frequency ($\geq 70\%$), low frequency ($\leq 30\%$), and medium frequency (30-70%), according to their consistency across replicates/samples. We initially assessed changes in compartment frequency between treatment groups or ATs.

To compare the compartment status between groups/ATs in more detail, we defined A/B switches and A/B variables using the following pipeline: first, we

defined a set of common A/B compartments (with >80% of individuals/replicates exhibiting the same chromatin status). The compartment status of a bin that exhibited opposite trends between groups/ATs was defined as an A/B compartment switch. We then identified regions with the same compartment status between groups/ATs, but with statistically significant differences in compartment scores (*i.e.*, the A-B index) between groups/tissues ($|\Delta AB \text{ index}| > 0.75$ and $q \text{ value} < 0.05$, Students' *t*-test and adjusted FDR), which were then designated as A/B variables.' (main text: page 23, lines 710-744)

Comment 1-15

167: "genes near compartments" => genes "in" compartments? How was the assignment made?

Response 1-15

Revised as suggested. We assigned a gene to a compartment, if the transcription start site of that gene was located within the 20-kb compartment bin.

'In contrast, genes in compartments that shifted between ATs or conditions exhibited higher changes in expression than those within common bins.' (main text: page 7, lines 172-174)

Comment 1-16

179: "the vast majority of which did not vary between replicates" => According to Fig S17b only a minority of them are in the High frequency category.

Response 1-16

Revised as suggested.

Comment 1-17

189: "genes near shifted TAD boundaries" => how is "near" defined?

Response 1-17

We have added a corresponding explanation to the Main text.

'genes near shifted TAD boundaries (*i.e.*, within ± 100 kb of a boundary)'. (main text: page 7, lines 195-196)

Comment 1-18

189: "genes near shifted TAD boundaries showed more dramatic changes in expression than those near the common boundaries" => This is not clear to me in Fig 2h, as most of the comparisons do not show any significant change actually. The author should rephrase and maybe highlight the contrast with results from A/B compartments.

Response 1-18

We have rephrased the sentence and added some text highlighting the relative difference between results of TAD boundaries and A/B compartments (see below). We appreciate this suggestion, since it has resulted in more precise interpretation of our results and provided some new insights worthy of discussion.

'Consequently, genes near shifted TAD boundaries (*i.e.*, within ± 100 kb of a boundary) in some of the comparisons showed slightly higher (though not significant) changes in expression than those near the common boundaries²⁹ (Fig. 2h). The influence of shifted boundaries on gene expression was relatively milder than that observed in A/B compartments. These differences in expression could potentially reflect the rewiring of the distal PEI networks caused by changes in the chromatin insulation status at these loci.' (main text: page 7, lines 195-201)

Comment 1-19

191: "These substantial differences" => same than above.

Response 1-19

We have revised the description.

Comment 1-20

197: "EN1 expression in ULB is higher than in GOM ($P = 5.30 \times 10^{-5}$, Wilcoxon rank-sum test)"

Since a DGE analysis was performed using a proper normalization method and FDR correction, I do not understand the point of using a Wilcoxon test on TPM values without adjusting p-values. See also line 205 for instance.

Response 1-20

According to your suggestion, we have changed these to adjusted p-values determined by edgeR.

'Likewise, *EN1* expression in ULB (TPM = 11.35) is higher than in GOM (TPM = 0.07) (**adjusted $P = 2.46 \times 10^{-13}$**) (**Fig. 2i**)' (**main text: page 8, lines 205-207**)

'However, this boundary is absent in GOM under NC conditions, and *TCF21* expression is largely increased (TPM = 6.22; **adjusted $P = 2.12 \times 10^{-11}$** ; **Fig. 2j**)' (**main text: page 8, lines 211-213**)

Comment 1-21

219: "PEIs were preferentially located within TADs"

Given that PEIs are on average much smaller than TADs, how did the authors tested if the proportion of intra-TAD PEIs was significantly higher than expected by chance? Similarly, how was the enrichment with CTCF loops tested? I could not find the details in Sup note 4.

Response 1-21

We have added a detailed explanation of this statistical test to the legend of **Supplementary Fig. 25**.

'To test the preference of PEIs located within TADs or CTCF-mediated loops, we generated random/spurious PEI data sets, according to the number and

length distribution of observed PEIs for each chromosome. Statistical significance was then assessed by Chi-square Test.' (Supplementary Information, page 40, lines 639-642)

Comment 1-22

228: Unlike all previously mentioned features, PEI usage does not seem to show a higher similarity between replicates within ATs vs. groups (~36% consistent PEIs in both cases, as supported by Fig S19a). Fig 3c-d and Fig S19c suggest otherwise. Could the authors comment on that?

Response 1-22

Thanks for this thoughtful comment. Based on your suggestion regarding PEI usage, we tried to explore patterns of PEI usage by evaluating the chromatin interactome across samples (**revised Supplementary Fig. 20a**), instead of through the overly stringent overlapping of genomic position by promoter and enhancers, as shown in **original Supplementary Fig. 19a**. t-SNE analysis revealed that PEI usage profiles were more similar among groups than they were between ATs (**revised Supplementary Fig. 20a and 20b**), which recapitulated the previously mentioned features at multiple levels, *i.e.*, expression (**Fig. 1g**), A/B compartment (**Fig. 1h**), IS values (**Fig. 1i**), as well as number of enhancers interacting with a promoter/gene (**Fig. 3c**), and RPS values (**Fig. 3d**).

We also revised our description of these results in the Results section. (**main text: page 9, lines 234-237**)

'We found that the chromatin interactome of PEIs was more consistent across the three treatment groups for a given AT than among different ATs within a single treatment group (**Supplementary Fig. 20a, b**)'

Revised Supplementary Fig. 20. Transcriptional regulation through chromatin rewiring of PEIs in different ATs across groups.

a t-SNE plot of the intensity of promoter and enhancer interactions (PEI).

b Violin plot of the distance between PEI interaction intensities between samples from the t-SNE plot in **a**.

Original Supplementary Fig. 19a. Sharing of PEIs between pairwise ATs and between treatment groups for each AT.

Comment 1-23

355: Fig S35 does not support that conclusion.

Response 1-23

Thanks for bringing this issue to our attention. We have made corresponding revisions to our interpretation of the results.

‘The global pattern of higher divergence between VATs and SAT compared to divergence within VATs was obscured by the relatively greater variation among human samples. However, the relatively higher divergence between inflammatory GOM and metabolic SAT (i.e., ASA) was still evident in RPS analysis and transcriptomic profiles, as well as in our analysis of local spatial

context (reflected by A-B index and IS values). Moreover, we also observed that visceral RAD samples were closer to SAT, distinguishable from the congeneric GOM and MAD, similar to our findings of ATs in a pig model (**Supplementary Fig. 37**).’ (main text: page 12, lines 359-367)

Comment 1-24

382: *"with rapidly evolving [...] than those" => a "more" might be missing*

Response 1-24

Revised as suggested.

Comment 1-25

433: *"supporting the 'ortholog conjecture'" => isn't it "questioning" more than supporting, as the orthologous conjecture seems to advocate for a higher conservation of orthologous vs. paralogous gene?*

Response 1-25

Thanks for pointing out this unintentional error. The reviewer is correct, and we have accordingly modified the text.

‘**This finding is consistent with accumulating evidence apparently contradicting the ‘ortholog conjecture’.**’ (main text: page 15, lines 444-445)

Comment 1-26

677: *How was the quantile normalization performed? Tool name (if not Juicer), version, parameters? Same question for the O/E transformation line 679.*

Response 1-26

We have added these details to the Methods section.

‘**For each autosome, the normalized observed contact matrices were generated using the Knight-Ruiz algorithm (to remove intrinsic biases within the matrix) in**

the Juicer toolkit with quantile normalization conducted using BNBC (version 1.0.0) (to remove biases between samples), both set to default parameters⁹⁷.’ (main text: page 23, lines 698-702)

‘We then transformed the normalized observed contact matrices into an observed/expected (O/E) matrix by dividing each normalized observed contact frequency by its corresponding expected contact frequency (calculated as the average observed contact frequency for all loci at a certain distance), using the publicly accessible script, generate.o.e.matrix.py (https://github.com/JiamanZhang/Lab_Porcine-Adiposes_paper_codes/tree/main/Lab_OE_matrix).’ (main text: page 23, lines 703-709)

Comment 1-27

692: "The A-B index was then calculated at a 20-kb resolution, as previously described"

Could the authors give more information about how the A-B index was computed exactly? Was this done using Juicer as mentioned above or with another tool?

Response 1-27

We have added these details to the Methods section.

‘PCA was performed to generate PC1 vectors for each autosome per sample at 100-kb resolution using the ‘prcomp’ function in R (version 3.6.1) with default parameters.’ (main text: page 24, lines 712-715)

‘Spearman’s coefficient r was calculated between PC1 and genomic characteristics, including gene density and GC content, for each autosome using the ‘cor’ function in R (version 3.6.1) with default parameters.’ (main text: page 24, lines 715-717)

‘The A-B index was then calculated at 20-kb resolution using the publicly available code, get_ABindex.py (https://github.com/JiamanZhang/Lab_Porcine-

Adiposes_paper_codes/tree/main/Lab_AB_compartment), as previously described⁹⁸. (main text: page 24, lines 720-724)

Comment 1-28

707: Which matrices were used to call TADs? O/E matrices were used for the A/B compartment prediction but I guess that TAD finding was performed on raw or KR-normalized matrices. This is not precised here. How IS and LBS were computed is not precised either.

More generally, could the authors make sure to precise for each analysis: input data, software name, version, and parameters?

Response 1-28

Thank you for pointing out this unclear point. As suggested, we have made revised and carefully checked all related issues throughout the entire Methods section.

'The deDoc⁵ (version 1.0.0) program at default parameters was used to identify TADs in the 20-kb normalized contact matrices (generated by the Knight-Ruiz algorithm and quantile method) for each autosome. TADs shorter than 100-kb were removed. To characterize the strength of each TAD structure, we calculated IS⁹⁹ (insulation score) using matrix2insulation.pl (<https://github.com/dekkerlab/cworld-dekker>) with the parameters (-v -is 260,000 -ids 200,000 -im mean -nt 0.1 -bmoe 0) and LBS²⁹ (local boundary score) using the get.samples.chr.LBS.value.py script (https://github.com/JiamanZhang/Lab_Porcine-Adiposes_paper_codes/tree/main/Lab_TAD_LBS/codes) with default parameters at 20-kb resolution.' (main text: page 25, lines 746-756)

Comment 1-29

747: "were identified by FIMO102, phastCons, and phyloP (downloaded from UCSC)"

Could the authors be more specific about where to access phastCons and phyloP information for pig in UCSC?

Response 1-29

We have added this information to the Methods.

‘We also characterized sequence elements embedded in regions of variable or stable A/B compartments or TAD boundaries, including CTCF motifs that were identified by FIMO¹⁰³ (version 5.1.1) with default parameters; phastCons and phyloP of pig were obtained by liftovering the corresponding values for the human genome version hg38 (downloaded from UCSC, <http://www.genome.ucsc.edu>) with pig reference genome Sscrofa11.1 using the UCSC LiftOver tool; housekeeping genes and transposon elements (TE) generated in a previous study⁶.’ (main text: page 26, lines 784-791)

Comment 1-30

894: How was CAFE used? Were the load/tree/lambda/report commands executed? Was a Newick phylogenetic tree provided, and if so, where was it obtained from?

Response 1-30

We have added this information to the Methods.

‘We used five species (humans, pigs, mice, rats, and cows) to identify expanded or contracted gene families using CAFE¹¹⁹ (<https://github.com/hahnlab/CAFE>) with default parameters. A phylogenetic tree of five species was obtained from the TimeTree database (<http://timetree.org>).’ (main text: page 31, lines 947-950)

Comment 1-31

905: When testing for functional enrichment of GO terms in the DE gene set, what was the background gene set? All the annotated genes or only the ones that were expressed in the ATs?

Response 1-31

All annotated genes in the genome were used as the enrichment background. In this study, Gene Ontology (GO) and KEGG pathway functional enrichment analyses were performed using Metascape (<http://metascape.org>), which included two options for the background gene sets: all annotated genes or a manual input gene set. As suggested by Metascape for most unbiased backgrounds, the precise list of background genes is often unknown, and as a result, the cutoff for generating the input hit list is somewhat subjective. Thus, the practice of using the whole genome for the analysis background is generally accepted. An approximate unbiased background gene set may shift the absolute values of the p-values, but we expect it has only limited effects on the relative ranking of the enriched terms. In the initial analysis, we independently tested both the total annotated genes and the significantly expressed genes in corresponding tissues to assess the biological relevance of output results.. Despite minor changes in significance values, no obvious differences were observed between these two backgrounds, especially in the relative ranking of the most significantly enriched terms (**Figure R2**). Therefore, we opted to use all annotated genes as the background gene set across the entire functional enrichment analysis.

Figure R2. Functional enrichment terms for highly expressed AT-specific genes identified in pairwise comparison between GOM and ULB. Functional enrichment

analysis was performed for genes specifically expressed in GOM **(a)** and ULB **(b)**, based on a background of total annotated genes (left) or significantly expressed genes (right).

Comment 1-32

1256: I do not understand the Circos plots in Fig 2a. Also, I cannot read what is written in the inner circles, the font is too small.

Response 1-32

Thank you for pointing out this issue. We have redrawn the Circos plots with a more legible font size and added more detail to the corresponding description in the figure legend.

a, b Differences in the frequency of A/B compartments in pairwise comparisons of ATs and treatments. Proportion plot showing stability and variability of compartment shifts across different adipose depots (a) and groups/treatments (b). The frequency of active compartment status was classified into three categories: high (H, red), medium (M, grey), and low (L, blue) frequency. Circos plots highlight remarkable changes from high-to-low or low-to-high frequencies between pairwise ATs in each treatment group (in 120° sectors) (a) or between treatment groups in each AT (in 90° sectors) (b). For Circos plots in (a), The outermost and second outermost circle display each VAT with 20kb A/B compartment

bins with different frequencies across individuals/replicates (classified as three High, Medium, or Low). The third circle shows the A/B bin frequencies for subcutaneous ULB. Red or blue vectors indicate changes in frequency from low-to-high or high-to-low, respectively, in pairwise comparisons between adipose tissues in adjacent rings; vector thickness is proportional to frequency of the changed compartment. The number of bins is shown in parentheses. For Circos plots in (b), the presentation is the same as that in (a), but circles from outermost to third represent WL, WG, and NC groups, respectively.'

Comment 1-33

Fig S4: How were HiCrep, Genomedisco and QuASAR used exactly and on which data: raw interaction matrices, ICE and/or O-E normalized matrices?

Response 1-33

Thanks for pointing this out. These softwares were used on the normalized observed contact matrices (at 100-kb) (generated by Knight-Ruiz algorithm and quantile method by BNBC), as described in the Methods. We have added information to the figure legend.

'**b** Comparison of variation in global genome architecture between groups (blue) and ATs (yellow) using HiCrep SCC (stratum-adjusted correlation coefficient), Genomedisco, and QuASAR based on 100-kb normalized contact matrix, generated by Knight-Ruiz and BNBC (see Methods section '*Initial processing of Hi-C data*'). (Supplementary Information: page 16, lines 343-346)

Comment 1-34

Fig S25a: "Enhancers"

Response 1-34

Thanks. We have corrected this error.

Reviewer #2**Comment 2-1**

This study has done very comprehensive investigation of potential regulatory mechanisms of obesity phenotypes in a pig model by primarily focusing on 3D genome chromatin interaction from anatomically distinct adipose tissues under different diet-induced condition. The results provide one of the largest high-resolution 3-D genome map in adipose tissues in farm animals and will be a great resource to identify potential genetic variants particularly in non-coding regions associated with obesity, an important human disease. The data generated could also provide many potential hypothesis that need be further investigated in the field.

Here are some specific comments:

Response 2-1:

We are very grateful for the reviewer's careful consideration and positive remarks in support of our study. We are keenly and profoundly aware of the potential value this study represents as a resource for the research community as a whole.

Comment 2-2

1. There were significant phenotype and gene expression difference between weight gain and weight loss group compared to control group. If chromatin interaction plays a significant role in regulating gene transcription, some specific example to demonstrate how A/B compartment, TAD difference affect gene regulation would be desirable as the phenotypic difference due to diet is the most important biology this study is trying to investigate underlying regulatory mechanisms.

Response 2-2

Thanks for this constructive suggestion. We completely agree that uncovering the regulatory mechanisms (especially those mediated by chromatin

architecture) underlying the phenotypic and transcriptional differences associated with weight gain and weight loss is a primary focus of this study that was somewhat obscured by findings of differences between adipose tissues. Indeed, we found multi-level remodeling of chromatin architecture that underpins transcriptomic divergence in ATs which is potentially linked to progressive metabolic risks in obesity development. For A/B compartmentalization, we found 27.13 Mb (~1.20% of the genome) regions of a given AT exhibited specific, distinct compartmental status or had significant differences in A-B index between nutritional conditions, which influences the expression of embedded genes (**Supplementary Note 2, page 7, lines 162-196**). For TADs, relatively few boundary changes were detected in each AT across treatment groups (average 0.18%, ~22 boundaries), which is consistent with previous findings of TAD stability across metabolic and physiological changes, and reflects a more fundamental organizational unit of chromosomes (**Supplementary Note 3, page 8, lines 199-234**). We also observed relatively fewer changes in CTCF-loops between nutritional conditions than between ATs (**Supplementary Note 4, page 9, lines 235-259; Supplementary Fig. 21c, d**). To emphasize and highlight these changes among treatment groups (*i.e.*, relevant to weight gain or weight loss), we added new figure panels (**revised Figure 1**), and made a new plot to better illustrate the hypertrophic (**revised Supplementary Fig. 4c**), hypoxia-related, and inflammatory genes (**revised Supplementary Fig. 15**) with changes in compartment status between the NC and WG groups.

j-l The proportional distribution of projection distance for t-SNE plots of gene expression in **g** (**j**), A-B index in **h** (**k**), and IS in **i** (**l**) between each dot and a given line ($y = kx$, $k = -0.3, -1, -5$ for gene expression, A-B index and IS, respectively) across groups.

Revised Supplementary Fig. 4. Pattern of compartment status and expression of genes suggesting characteristics of distinct ATs.

a Compartment status and expression pattern of hypertrophy genes ($n = 10$) in each adipose depot across weight gain and loss. Statistical significance was determined using

Wilcoxon rank-sum test. *P* value is shown above each box of each AT.

b Histogram showing the compartment status and transcription of representative hypertrophy gene (i.e., *TM4SF1*). The data are shown as mean with SD (NC *n*=12, WG *n*=46). Statistical significance was determined using a Wilcoxon rank-sum test.

c A representative gene (*TM4SF1*) with changes in A/B compartment status during weight gain for each adipose depot. Tracks show compartment status (top) and gene expression level (bottom).

d Histogram showing the compartment status and transcription of representative hyperplasia genes (i.e., *PPARG*). Data are presented as mean values \pm SD (*n* number is listed above each bar). Statistical significance was determined using a Wilcoxon rank-sum test.

e Heatmap showing the pattern of (left panel) compartment status and (right panel) expression of *HOX* genes across adipose depots, in which *HOXD* clusters showed a remarkable difference in compartment status between SAT and VATs. Within each treatment, the compartment status of all nine *HOXD* cluster genes (typically, *HOXD4*, a key regulator in controlling the adipocyte development [i.e., pre-adipocyte differentiation] of SATs) are similar in ULB and RAD, which are more active than in GOM and MAD (for example, A-B scores in NC, ULB: 2.37, RAD: 2.14, GOM: 1.15, MAD: 1.69).

f Change of compartment status of 55 hyperplasia genes between adipose depots in each group.

g Expression level of 13 mitochondrial protein-coding genes in each AT between NC and WG group. Statistical significance was determined using a Wilcoxon rank-sum test.

h t-SNE plots based on the compartment status of 37 *HOX* genes. The ellipses indicate the samples of each AT with similar profiles, constructed at a probability of 0.85.

Revised Supplementary Fig. 15. Histograms of compartment status and transcription level of representative hypoxia genes (i.e., *EGR1*), as well as representative inflammation genes (i.e., *CD14* and *TNFSF12*) in each AT after weight gain. The data show means with SD (NC $n = 12$, WG $n = 46$). Statistical significance was determined using a Wilcoxon rank-sum test. Tracks show compartment status (top) and gene expression level (bottom).

Comment 2-3

2. The statement in Line 100-101 "all substantially diverged in the WG and WL groups compared to the NC group" in Fig 1g-i is not accurate (at least from figures presented, a different PCA plot only using NC, WG and WL maybe needed for this purpose).

Response 2-3

Thank you for pointing out this issue. We have revised this sentence, and added a new panel (**new Fig. 1 j-l, see details in response 2-2**) to better illustrate differences among nutritional treatment groups.

'Total RNA-seq for the corresponding AT samples indicated that the transcription, compartmental rearrangements, and local spatial context (reflected by insulation scores [IS]) **diverged** in the WG and WL groups compared to the NC group (**Fig. 1g-l**), suggesting that both excess and insufficient caloric intake can reshape chromatin architecture and transcriptomic patterns in ATs'. (**main text: page 4, lines 103-108**)

Comment 2-4

3. Line 144-146, meaning is not clear.

Response 2-4

Thank you for pointing out this. We have revised this sentence, and added statistical analysis in revised **Supplementary Fig. 8**.

'Using the distance between samples in the t-SNE as a metric, we found that the average distance between replicates was slightly less than that between treatment groups (Wilcoxon rank-sum test, $P < 1.96 \times 10^{-9}$), and markedly less than that between pairwise ATs (Wilcoxon rank-sum test, $P < 2.2 \times 10^{-16}$) (**Supplementary Fig. 8**), suggesting considerable intra-group heterogeneity.' (**main text: page 6, lines 148-152**)

Comment 2-5

4. Line 157 "most A/B compartments were invariable within replicates" does this mean among individuals pigs within each treated group and within each tissue?

Response 2-5

Yes, you are correct. It means most A/B compartments for each tissue did not vary among individual pigs within each treatment group.

Comment 2-6

5. For Fig. 2b, it seems RPS score could be high at 20-30, why the range of RPS is only up to 5?

Response 2-6

Thanks for pointing out this issue in our data presentation. Actually, the X axis is not the range of RPS value. As described in figure legend, genes with RPS>0 were divided into five percentiles. For clarity, we have revised **Fig. 3b**.

Revised Fig. 3b Positive correlation between gene expression and RPS. Genes with RPS > 0 were divided equally into five percentiles. Data show medians \pm SD, and dots represent the value of each AT across groups. RPS: a regulatory potential score for each gene.

Comment 2-7

6. Fig. 2e, it seems there is no positive correlation between TPM and RPS as presented in Fig 2b between RAD and ULB (totally negative correlated). In addition, the interaction between promoter and activities of enhancers interacting with promoters seemed not reflect the RPS score in the figure. Again, is there any example to compare between different treatment groups as they are more biologically important.

Response 2-7

Thank you for pointing out this inadvertent omission. We carefully checked the data and found that the RPS for ULB and RAD are accidentally reversed. We have corrected this issue.

We agree that comparisons between treatment groups are more biologically important. Indeed, we discuss this topic in genes with covariation between RPS and gene expression (**page 10, lines 268-292**), and the results are shown in **Fig. 3 j-l**, and **Supplementary Fig. 32**. According to your suggestion, we now present more information for each example of differences between treatment groups, such as in **revised Supplementary Fig. 4c** and **revised Supplementary Fig. 15 (see more details in response 2-2)**. To better address these concerns and make the data more readily accessible to readers, we have also added more details in new **Supplementary Data 2** and **Supplementary Data 3** and **5**.

Comment 2-8

7. For Fig. 3i, for most genes, RPS is positively correlated with TPM, but for some genes, totally opposite, any explanation on this (C5AR1, CXCL8, HCK etc.)? it would be great to provide a figure for some of genes with TPM from low to high or high to low in the order of WG, NC, WL to demonstrate how A/B and TAD information in respective tissue, treatment groups, their conservation information across multiple mammals in potential promoters and its interacting enhancers contribute to gene regulation.

Response 2-8

Yes, the figure indeed shows that not all genes have a positive correlation between TPM and RPS. Gene expression can be coordinately regulated by various pre- and/or post-transcriptional modifications or mechanisms, such as DNA methylation, active or inactive Histone modifications, promoter-enhancer interactions (a focus of this work), as well as microRNA-mediated RNA silencing. Thus, a potential explanation for this ambiguous trend is that individual genes are expressed to varying degrees in the contexts of their diverse cellular functions, and a subset of regulatory elements or various mediators may be better described as fine-tuning rather than independently inducing or silencing transcription. Multiple regulatory interactions or mediators can also exert synergistic or non-linear effects on gene regulation⁷.

As suggested, we show the set 15 genes with higher RPS and expression after weight loss (i.e., WL versus WG groups) in a separate panel (**revised Fig. 3i**).

Revised Fig. 3i. Heatmaps of RPS (left) and expression (right) patterns of 15 inflammation-related genes that are highly expressed in the WG group compared to the NC group and remained stable in the WL group. Differential RPS genes: genes with changes in RPS FC [fold change] > 1.5, $|\Delta| > 2$; Otherwise, Non-differential RPS genes.

As suggested by the reviewer in Comments **2-1** and **2-6**, we have added Figures (**revised Supplementary Fig. 4c** and **revised Supplementary Fig. 15**) better illustrating A/B compartment arrangement of the hypertrophic, hypoxia-related, and inflammatory genes between the NC and WG groups. As shown in **Fig. 3h, i** and **revised Supplementary Fig. 32**, we also present the expression patterns and RPS of genes that are highly expressed in the WG group compared to the NC group which remained stable in the WL group.

We have also added an analysis of promoters and interacting enhancer conservation across multiple mammals (30-way phastCons value) in **revised Supplementary Fig. 32** to further assess the conservation of these regulatory elements across species or uncover their functional roles in well-established model systems such as mice.

Revised Supplementary Fig. 32. Expression patterns and RPS of candidate obesogenic memory genes.

a–d Schematic representation of PEIs for typical genes, including *SHARPIN* in GOM (**a**), *CD180* in MAD (**b**), *TNFRSF1A* in RAD (**c**), and *CLECT7A* in ULB (**d**) that show highly expressed obesogenic memory established in the WG group compared to NC group which remains stable in the WL group. From left to right: Hi-C maps indicating promoter-centered interactions (upper left) and gene expression levels (lower left). Interaction metaplots of

promoter-centered regions and 3D structural models (middle). Promoters: blue spheres; low-activity enhancers: green spheres; moderate-activity enhancers: pink spheres; high-activity enhancers: purple spheres; and PEIs: connecting lines. Difference in PEI intensity between pairwise groups comparisons (right).

e-f The conservation information across multiple mammals (30-way phastCons value) in the promoters and its interacting enhancers for *SHARPIN* in GOM, *CD180* in MAD, *TNFRDF1A* in RAD, *CLECTA* in ULB during weight gain or loss. The tracks show conservation information alongside the designated 5-kb regions with 5-kb up- and downstream regions. The shadow box shows the 5-kb region of promoters (blue) or enhancers (grey). Enhancers/promoter are presented from top to bottom based on their linear genomic position, corresponding to **a-d**, respectively.

Reviewer #3**Comment 3-1**

Authors used a pig model to analyze the multi-scale structural dynamics of the 3D genome. They found the dramatic transcriptomic and chromatin architectural changes among the four ATs under different nutritional conditions. Then they revealed high similarity in the regulatory circuitry of genes responsible for the obesity phenotype and identified non-conserved elements in species-specific gene sets that underpin AT specialization. They also provide a tool for discovering obesity-related regulatory elements. Summary, this research provides transcriptomic and chromatin architecture of pig as a model system and compares it to other species. They confirmed the phenotype and underlying mechanisms of obesity development in humans and rodents. This study is interesting and well designed, however there are still some problems, which need to be solved before it is considered for publication.

Response 3-1:

We are very grateful for the reviewer's careful consideration and positive remarks in support of our study.

Comment 3-2

1. The weight loss pig was getting from the weight gain pig rather than from the normal pig. How about the obesogenic and metabolic memory mechanism of getting weight loss from normal pigs?

Response 3-2

Thanks for this valuable and highly interesting question. Actually, we explored the phenomenon of 'obesogenic memory', defined as a long-term increase in AT inflammation and insulin resistance due to high fat diet-induced obesity that persists even after weight loss by caloric restriction.^{8,9} Therefore, we analyzed inflammatory genes that conform to criteria of increased expression/activity in the weight gain group compared to that in control animals which retained high

expression levels in weight loss group (*i.e.*, comparable expression to WG, and higher than that in the NC group) (**revised Fig. 3h, i**).

Given that adipose tissue undergoes morphological and functional changes during obesity and represents one of the largest immunologically or inflammatory active tissues, our goal was to investigate whether adipose tissue from formerly obese animals displayed any residual transcriptional regulatory effects that could potentially contribute to disease. Other recent studies have reported this phenomenon in mice¹⁰. Briefly, these studies found that despite normalization of body weight and correction of metabolic abnormalities after weight loss, adipose tissue (specially visceral adipose tissue) in formerly obese mice retains pathological properties, such as typical crown-like structures, such as formation of a syncytium of macrophages around dead adipocytes. The results in our pig model thus provide resource for comparative analysis across various animal models regarding this topic, and especially focus on the regulatory role of chromatin architecture in obesity- and obesogenic memory-related transcriptional changes. By contrast, regulatory changes associated with weight loss in normal (non-obese) animals are not considered obesogenic memory and, although interesting, fall outside the scope of our current work.

To improve clarity on this point, we have modified our statement as follows:

‘Consistent with this hypothesis, five markers of ‘obesogenic memory’ in mice (TNF, IL6, IL10, CCL2, and CCL3) retained high expression and similar pattern of interactions in promoter-centered regions (and persistent AT inflammation) following weight loss (**compared to that in WG group**) in our pig model’. (**main text: page 10, lines 277-281**)

Comment 3-3

2. In this study, some Hi-C and transcriptome data were downloaded from the public platform. So whether they come from the same individual? If not, there will be large bias in the integration analysis due to experimental conditions and individual differences.

Response 3-3

Thanks for pointing out this issue. Yes, among the 249 samples with Hi-C and transcriptome data used in this study, data from GOM and ULB tissue for nine samples in the NC group were downloaded from the SRA database and were published in two studies by our group (Zhang et al¹¹, *Journal of Animal Science and Biotechnology*, 2022; Jin et al¹², *Nature Communications*, 2021) (as registered in **Supplementary Data 1**).

These downloaded Hi-C and transcriptome data were obtained from seven animals in the NC group and belonged to the same experimental population, with highly comparably body weights and health status. Moreover, the *in situ* Hi-C assays and RNA sequencing of these samples were conducted in the same batch and conditions. Thus, we opted to use these data based on the expectation of minimal or no bias or batch effect.

Comment 3-4

3. Whether there are batch effects in Hi-C data is not discussed. If so, how to conduct batch removal? The paper needs to explore the batch effect of Hi-C data sampled from different batches.

Response 3-4

Thanks for raising this question. We are confident that we have minimized or eliminated potential batch effects that could have been introduced by variability in technical aspects or protocols in the laboratory.

First, for this study, we conducted *in situ* Hi-C assays for all 249 adipose tissue samples in experiments that spanned months and were conducted by the same, highly proficient technicians to guarantee the highest possible consistency in experimental conditions, sample procurement, sample storage, library preparation, and sequencing towards the specific goal of minimizing batch effects.

Furthermore, we employed BNBC, an accepted method for normalization and batch correction of Hi-C data that can substantially improve comparisons across samples (see **reference 97**). This method could also help eliminate sources of technical or artefactual (i.e., not biologically relevant) variation.

Indeed, we observed no obvious abnormal patterns or phenomena suggesting batch effects in our final normalized observed contact matrix. As shown in **new Supplementary Fig. 2, Fig. 1g-i, and Supplementary Fig. 8**, we found the expected global patterns, that is, higher similarity/correlation between replicates than between treatment groups, and higher similarity between treatment groups than between ATs.

Therefore, we are confident that no potential *a priori* batch effects affected current results. We thank again the reviewer for ensuring the rigor of our analysis.

New Supplementary Fig. 2. The distribution of A-B index (a), IS (b), TAD size (kb) (c), and CTCF loop size (kb) (d) proportions across groups for a given tissue. Colored lines represent mean value across replicates and shading around the mean represents standard deviation across replicates.

Fig. 1g–i Comparison of variation in gene transcription (g), AB compartment (h), and IS (i) between adipose depots and between groups. t-distributed stochastic neighbor embedding (t-SNE) clustering of samples. In t-SNE plots, ellipses indicate AT samples with similar profiles, constructed at a probability of 0.85.

Revised Supplementary Fig. 8. Correlation of gene expression, AB compartmentalization, and local spatial context (IS value) for pairs of samples between replicates, between tissues, and between groups. **Statistical significance (P values) was determined using Wilcoxon rank-sum test. A-B index: between replicates vs. between groups, $P < 2.2 \times 10^{-16}$; between replicates vs. between ATs, $P < 2.2 \times 10^{-16}$; IS value: between replicates vs. between groups, $P < 2.2 \times 10^{-16}$; between replicates vs. between ATs, $P < 2.2 \times 10^{-16}$; Expression: between replicates vs. between groups, $P = 1.96 \times 10^{-9}$; between replicates vs. between ATs, $P < 2.2 \times 10^{-16}$.**

Comment 3-5

4. 6. Section "Gene duplication accompanied by rapidly evolving enhancers" mentions that there are many of specific expanding genes families in human, which are correspondingly contracted in porcine genomes. The effects of the contractions on pigs should be discussed.

Response 3-5

Thanks for your constructive suggestion. In the Results section '**Gene duplication accompanied by rapidly evolving enhancers**', we aim to investigate the potential evolutionary regime of enhancers (regulatory elements) that accompanies the expansion or contraction of gene families. As described in the Methods, we used CAFE with five species (humans, pigs, mice, rats, and cows) to identify expanded or contracted gene families. Genes with copy numbers (the number of homologs, i.e., gene IDs, for a given gene within a species, collectively termed a gene family) that were greater (or lower) in humans than the average of the other four species and "Viterbi P -values" < 0.05 calculated by CAFE, were considered expanded or contracted genes in humans.

To address the reviewer’s concerns about expanding gene families (72 gene families as shown in **Fig. 6a**) in humans that may have contracted in pigs, we surveyed 32 such gene families that were specifically contracted in pig (**Fig. 6a**) and found that none of these had expanded in humans (**Table R1**). Thus, no effects of contraction can be identified in pig that correspond to gene family expansion in humans. We again thank the reviewer for this stimulating question.

Table R1. List of expanded gene families in human and contracted families in pig.

Significantly expanded gene families ID in human (72)	Significantly contracted gene families ID in pig (32)
family14, family39, family41, family58, family62, family455, family623, family626, family690, family762, family806, family1077, family1089, family1321, family1415, family1431, family1814, family1833, family2345, family2472, family2523, family2805, family3469, family3776, family3835, family4280, family4406, family4578, family4660, family4883, family4975, family5032, family5459, family5531, family5704, family6016, family6366, family6389, family6521, family6601, family6652, family7193, family7209, family7225, family7670, family8023, family8098, family8408, family8655, family9211, family9922, family10048, family10969, family10976, family11043, family11231, family11315, family12077, family12283, family12327, family13145, family13959, family14334, family14514, family14565, family14672, family15553, family17262, family17608, family17785, family18165, family18486	family484, family1417, family1478, family1556, family2472, family2995, family3559, family3835, family4331, family4578, family5665, family6155, family6378, family6389, family8081, family8100, family8487, family9156, family9755, family11490, family12143, family12168, family12371, family12380, family12384, family13451, family14012, family14334, family14367, family15254, family17204, family18166

Comment 3-6

5. In fig.1g-i, t-SNE clustering of different treatment groups is not clearly distinguished.

Response 3-6

As noted by reviewer 1 (**see Response 1-4**), we indeed observed objectively larger differences between ATs than between groups, which comports well with

the dramatic regulatory differences associated with different cellular function of ATs compared to relatively mild regulatory differences associated with various physiological/nutritional treatments. Although the changes in body weight and morphology associated with nutritional treatment were indeed dramatic, obesity is a relatively slight physiological alterations compared to that in malignant cancer⁵ or cell differentiation⁶, for example. Based on the reviewers' suggestions, we have revised our data presentation to better illustrate the differences between groups (**revised Fig. 1**).

Fig. 1j-l The proportional distribution of projection distance for t-SNE plots of gene expression in **g** (**j**), A-B index in **h** (**k**), and IS in **i** (**l**) between each dot and a given line ($y = kx$, $k = -0.3, -1, -5$ for gene expression, A-B index and IS, respectively) across groups.

Comment 3-7

6. In section "Rewiring of PEIs underpins the AT transcriptional program response to different nutritional conditions", the detail definition and quantitative characteristics of "Rewiring of PEIs" should be added.

Response 3-7

We appreciate this constructive suggestion. Actually, "rewiring of PEIs" is a general description of changes in connections between enhancers and promoters. In this study, to investigate PEI rewiring between ATs or groups, we calculated a regulatory potential score (RPS) for each gene based on the hypothesis that an enhancer's quantitative effect on a gene could depend on their spatial proximity. To this end, RPS was calculated using the following equation:

$$RPS = \sum_{i=1}^n \log_{10}(l_n) \quad (2)$$

in which n represents the number of enhancers linked to a gene in the aforementioned high confidence PEIs and l_n represents the normalized interactions (*i.e.*, the observed contact frequency minus the expected contact frequency) for each PEI (see Methods section '***Identification and analysis of Promoter-enhancer interactions (PEIs)***').

According to your suggestion, we have added further description to the Methods to improve clarity on this point.

'To investigate PEI rewiring, we compared the RPS between ATs/groups; differential RPS genes were defined as those genes with FC [fold change] > 1.5, $|\Delta| > 2$.' (main text: page 29, lines 874-876)

Comment 3-8

7. Is promoter bias excluded when using ROSE to find enhancers? It is recommended to set the -t parameter to remove the promoter region.

Response 3-8

Yes, the promoter region was excluded when identifying PEIs. As suggested, we have added more details of this analysis to the Methods.

'We used ROSE with a -t parameter set to 2000 to exclude promoter regions.' (main text: page 29, lines 881-882)

Comment 3-9

8. In line 825, I wonder what the O-E stands for here. It's seemed that the paper does not explain this abbreviation.

Response 3-9

We have changed the O-E abbreviation to a detailed description in the Methods.

'.....in which n represents the number of enhancers linked to a gene in the aforementioned high confidence PEIs and l_n represents the normalized

interactions (*i.e.*, the observed contact frequency minus the expected contact frequency) for each PEI'. (main text: page 28, lines 868-871)

Comment 3-10

9. In line 824, how to define the number of enhancers linked to a gene?

Response 3-10

Before we calculated the regulatory potential score (RPS) for each gene, we identified the putative enhancers (5-kb in length) for each gene, defined by over-represented interactions (high confidence PEIs with FDR values < 0.001 and interaction distance \geq 40 kb) with a given promoter region (5-kb in length) identified in the 5-kb resolution normalized contact matrices using the PSYCHIC algorithm¹³.

PSYCHIC is a computational model for analyzing Hi-C data to identify enriched DNA–DNA interactions¹³. In our study, we used this analysis to identify an average of 42,736 enhancers assigned to 10,602 promoters and a median bridging distance of ~131 kb. These putative enhancers were preferentially located within TADs or CTCF-mediated loops (**Supplementary Fig. 25c, d**), and multiple enhancers had additive effects on target gene transcription (**Supplementary Fig. 20c**), suggesting accurate predictions. We have modified our description in the Methods to hopefully improve clarity (**main text: page 28, lines 868-869**).

'in which n represents the number of enhancers linked to a gene in the aforementioned high confidence PEIs'

Reviewer #4

Comment 4-1

The authors investigated transcriptome and chromatin architecture in two types of adipose tissues (ATs) - subcutaneous and visceral (from 3 different depots), using an adult miniature pig as model organism for human obesity. Novelty of this study is limited, since number of papers have been published about transcriptome of porcine ATs as well as chromatin organization in ATs.

Response 4-1

Appreciate this opportunity to further examine the literature on this exciting topic. In response to the issue of novelty, we surveyed the current body of published studies on chromatin architecture in adipose tissues or adipocytes that use Hi-C, promoter capture Hi-C (pCHiC), or other advanced experimental procedures in the Google scholar and PubMed databases. To our surprise, only limited a number of studies^{11,14-19} (seven total papers, see **Table R2**) were published from 2017 to 2022, five of which were performed in humans, one in mice, and one in pigs.

Table R2. List of recent papers related to chromatin architecture in adipose tissues or adipocytes.

Species	Tissue resource	Experimental procedures	Title	Year	Journals
Human	Primary fat cells	pCHiC	Long-range chromosomal interactions increase and mark repressed gene expression during adipogenesis	2022	Epigenetics
Human	Fat tissue	pCHiC	A Compendium of Promoter-Centered Long-Range Chromatin Interactions in the Human Genome	2020	Nature Genetics

Human	Lipid-challenged human adipocytes	pCHiC	Reverse gene-environment interaction approach to identify variants influencing body-mass index in humans	2019	Nature Metabolism
Human	Primary human white adipocytes	pCHiC	Integration of human adipocyte chromosomal interactions with adipose gene expression prioritizes obesity-related genes from GWAS	2018	Nature Communications
Human	Human primary adipose stem cells	Hi-C	Long-range interactions between topologically associating domains shape the four-dimensional genome during differentiation	2019	Nature Genetics
Mouse	3T3-L1 preadipocytes	Hi-C and pCHiC	Dynamic Rewiring of Promoter-Anchored Chromatin Loops during Adipocyte Differentiation	2017	Molecular Cell
Pig	Adipose tissue	Hi-C	Reorganization of 3D genome architecture across wild boar and Bama pig adipose tissues	2022	Journal of Animal Science and Biotechnology

Comment 4-2

It is quite strange that Authors have not cited previous papers on this topic (e.g. DOI: 10.1186/s40104-022-00679-2). A new aspect of the study is comparison of chromatin architecture in adipose tissues from animals on different diets – normal diet (NC) as a control, a high-fat diet (WG) and calorie-restricted diet (WL).

Response 4-2

Thank you for pointing out this unintentional oversight. We now cite this relevant paper (which is one of our own pilot studies examining differences in chromatin organization of GOM and ULB adipose tissues between two pig breeds) in the revised Introduction section.

‘Three-dimensional (3D) chromatin architecture is a fundamental regulator of transcription⁷, and is organized in multi-scale hierarchical layers, including chromosome territories, compartments⁸, topologically associating domains (TADs)⁹, chromatin loops¹⁰, and long-range interactions between promoters and enhancers (PEIs)¹¹. **An earlier study of chromatin architecture in adipose tissues provided early clues highlighting the regulatory importance of chromatin organization in adipogenesis¹².** Nonetheless, a panoramic view does not illustrate the dynamic changes in chromatin architecture that underpin transcriptomic divergence in ATs that are potentially linked to progressive metabolic risks in obesity development and dietary interventions.’ (**main text: page 2, lines 46-55**)

Comment 4-3

However, design of nutritional experiment has many failures. The WL group was created from WG group and the group of animals was subjected to nutritional regimen for next 12 weeks. The main weakness of such study design is different age of animals/time of keeping animals, which can significantly affect genome functioning.

Response 4-3

In designing this dietary intervention experiment in pigs, we very carefully considered the relatively long duration of these experiments and the potential time-related effects on the targeted adipose tissues and body weight. To minimize unwanted effects, we employed two-year-old animals, approximately corresponding to thirty-years-old in humans²⁰, as representative of fully mature adult stage mammals (**Figure R3**). Generally, at this stage, the developmental and growth profiles have already plateaued, and thus body weight, fat mass, and phenotypic features of adipose tissues (such as size or number in

adipocytes) should remain relatively constant or stable in the absence of nutritional modification such as overnutrition or caloric restriction²¹. More specifically, in mammals, adipocyte number (*i.e.*, hyperplasia, a major determinant of fat mass in adults) undergoes a period of increase that lasts through adolescence, and ultimately determines the total number of adipocytes that the individual will have as an adult²¹.

Collectively, we are confident that the 12-week difference in age between WG and WL animals does not significantly affect phenotypic or genomic functional features of the adipose tissue examined in this work. Thus, the currently observed differences/changes in transcription and chromatin architecture between WG and WL group are mostly induced by nutritional intervention. Moreover, this experimental design allows us to examine the phenomenon of obesogenic memory, which, by definition, requires an obese state prior to weight loss. We again thank the reviewer for the opportunity to clarify these points.

[redacted]

Figure R3. Comparison of growth curves in human and micro-mini pig. The growth curves for height, thymus development, and hematogenesis in humans and micro-mini pigs are depicted above. Micro-mini pigs take approximately 12 months after birth to reach their fully mature body weight. In addition, epiphyseal lines are closed at 20 months of age in

micro-mini pigs, while in humans, they close at 15–17 years of age. The blue vertical line indicates the age of pigs used in this study. Figure was adapted from previous study²⁰.

Comment 4-4

There is also no explanation why the 3 groups differ in size.

Response 4-4

It is well known that the physiology of large animal models, such as dog, pig and especially non-human primates, closely resembles human physiology. However, the trade-off is that these species have high maintenance costs and especially long life cycles²². Taking these factors into account, we believe that the number of individuals in each treatment group is sufficient for robust transcriptome and 3D genome Omics analyses.

In the initial design of these animal experiments, all 68 individual pigs that were sampled for sequencing were selected from a larger population of pigs to ensure relatively conformance in body weight and health status prior to nutritional intervention. In particular, for the weight gain/loss groups, pigs were fed with a high-fat diet for a relatively protracted period of 22 weeks. Considering the potential unpredictable influence of feed intake bias across individuals, and the complexity of the obesity phenotype^{23,24} (e.g., variability in body fat distribution), we employed more pigs in the HFD group, finally obtaining fifty-six obese pigs that showed relatively comparable body weight after overnutrition (**Fig. 1b**). For pigs in the WL group subjected to caloric restriction (10% of the daily caloric intake of the normal diet), food intake was easy to control, and thus only ten individual pigs were required to observe an accordant reduction in body weight (**Fig. 1b**). Hopefully the reviewer understands that these decisions about sample size were not arrived at lightly or without careful consideration among our research team.

Comment 4-5

Application of two-tailed t-test for comparison (Fig. 1), when 3 groups are object of interest is not justified.

Response 4-5

We thank the reviewer for this constructive insight. We have re-analyzed these data using a Wilcoxon rank-sum test.

Comment 4-6

Differences in phenotype (body weight, adipocytes size et) are visible only when compare WL with WG, no NC. All other analysis (transcriptome and chromatin studies) are referenced to the NC. Since, most of the presented results concerns different nutritional conditions, improperly conducted nutritional experiment, may affect obtained data.

Response 4-6

In this study, pigs fed with HFD (*i.e.*, WG group) became markedly obese after 22 weeks compared to pigs fed a normal diet (*i.e.*, NC group). Ten of these obese pigs were then subjected to 12 weeks of extreme caloric restriction, resulting in a dramatic reduction in body weight. We believe the significant changes between NC and WG groups, but relatively little apparent difference in weight between WL and NC groups are both reasonable given the time frames for weight gain and weight loss, and the dramatic changes in weights from NC to WG, then WG to WL.

In all analyses, including transcriptome and each hierarchy of chromatin architecture, comparative analysis was performed between NC and WG group, or between WG and WL groups (see **Supplementary Fig. 3, 14, 23 and 27**), to investigate the differences between these states.

Comment 4-7

There are many other comments to the manuscript, e.g. the title is not appropriate - what the word “dynamics” is supposed to mean - it would have to refer to the variability over time, which had no place in this work;

Response 4-7

As suggested, we have changed the title,

‘Dynamic chromatin architecture of the porcine adipose tissues with weight gain and loss.’

Comment 4-8

the abstract is not very uninformative regarding the results obtained;

Response 4-8

We have rewritten the abstract. (**main text: page 1, lines 4-23**)

‘To identify the regulatory mechanisms of three-dimensional (3D) genome architecture underlying obesity phenotypes in anatomically distinct adipose tissues (ATs), we used an adult **female** miniature pig model with diet-induced weight gain/weight loss to generate 249 high-resolution *in situ* Hi-C chromatin contact maps of subcutaneous AT and three visceral ATs. **Investigation of transcriptomic and chromatin architectural changes among the four ATs under different nutritional treatments showed multi-level remodeling of chromatin architecture that underpins transcriptomic divergence in ATs. These changes are potentially linked to progressive metabolic risks in obesity development (e.g., increasing inflammation) and the existence of persistent obesogenic memory even after caloric restriction-induced weight loss.** Analysis of chromatin architecture among subcutaneous ATs of different mammals suggested the presence of transcriptional regulatory divergence that could be responsible for phenotypic, physiological, and functional differences in ATs. Analysis of regulatory element (enhancer) conservation in all four ATs in pigs and humans revealed **similarities** in the regulatory circuitry of genes responsible for the obesity phenotype and identified non-conserved elements in species-specific

gene sets that underpin AT specialization. This work provides an integrated, data-rich tool for discovering obesity-related regulatory elements through comparison of the 3D genome architecture of humans and pigs.'

Comment 4-9

in the introduction part no previous work done on pigs has been cited, etc.

Response 4-9

We have cited previous work in pigs.

Comment 4-10

Concluding, due to methodological weakness of the study, the manuscript cannot be recommended for publication in present form.

Response 4-10

We have carefully gone through the reviewers' comments in detail and believe that we have fully addressed these questions and concerns, especially for the concerns about the age effect and biological replicates in experiment design. We think the methodology in current study is appropriate, especially for studying the complex quantitative trait of obesity. We feel that our current work represents a breakthrough in this field that will provide a major resource for future research that will benefit our own studies as well as that of our competitors. While we do not wish to indiscriminately cast aspersions, we feel that this study was carefully planned and involved the work of a large team of talented individuals using state of the art analytical methods, which is reflected in the comments of other reviewers. We appreciate this reviewer giving us the opportunity to respond, but otherwise do not consider their recommendation of rejection to be rational or reasonable for our study.

References

- 1 Li, A. et al. Decoding topologically associating domains with ultra-low resolution Hi-C data by graph structural entropy. *Nat. Commun.* **9**, 3265 (2018).
- 2 Rao, S. S. et al. A 3D map of the human genome at kilobase resolution reveals principles of chromatin looping. *Cell* **159**, 1665–1680 (2014).
- 3 Sárvári, A. K. et al. Plasticity of epididymal adipose tissue in response to diet-induced obesity at single-nucleus resolution. *Cell Metab.* **33**, 437–453 (2021).
- 4 Tchkonja, T. et al. Mechanisms and metabolic implications of regional differences among fat depots. *Cell Metab.* **17**, 644–656 (2013).
- 5 Wu, P. et al. 3D genome of multiple myeloma reveals spatial genome disorganization associated with copy number variations. *Nat. Commun.* **8**, 1937 (2017).
- 6 Dixon, J. R. et al. Chromatin architecture reorganization during stem cell differentiation. *Nature* **518**, 331–336 (2015).
- 7 Song, M. et al. Cell-type-specific 3D epigenomes in the developing human cortex. *Nature* **587**, 644–649 (2020).
- 8 Schmitz, J. et al. Obesogenic memory can confer long-term increases in adipose tissue but not liver inflammation and insulin resistance after weight loss. *Mol. Metab.* **5**, 328–339 (2016).
- 9 Reilly, S. M. & Saltiel, A. R. Adapting to obesity with adipose tissue inflammation. *Nat. Rev. Endocrinol.* **13**, 633–643 (2017).
- 10 Hata, M. et al. Past history of obesity triggers persistent epigenetic changes in innate immunity and exacerbates neuroinflammation. *Science* **379**, 45–62 (2023).
- 11 Zhang, J. et al. Reorganization of 3D genome architecture across wild boar and Bama pig adipose tissues. *J. Anim. Sci. Biotechnol.* **13**, 32 (2022).
- 12 Jin, L. et al. A pig BodyMap transcriptome reveals diverse tissue physiologies and evolutionary dynamics of transcription. *Nat. Commun.* **12**, 3715 (2021).
- 13 Ron, G., Globerson, Y., Moran, D. & Kaplan, T. Promoter-enhancer interactions identified from Hi-C data using probabilistic models and hierarchical topological domains. *Nat. Commun.* **8**, 2237 (2017).
- 14 Garske, K. M. et al. Long-range chromosomal interactions increase and mark repressed gene expression during adipogenesis. *Epigenetics* **17**, 1849–1862 (2022).
- 15 Jung, I. et al. A compendium of promoter-centered long-range chromatin interactions in the human genome. *Nat. Genet.* **51**, 1442–1449 (2019).
- 16 Garske, K. M. et al. Reverse gene–environment interaction approach to identify variants influencing body-mass index in humans. *Nat. Metab.* **1**, 630–642 (2019).
- 17 Pan, D. Z. et al. Integration of human adipocyte chromosomal interactions with

adipose gene expression prioritizes obesity-related genes from GWAS. *Nat. Commun.* **9**, 1512 (2018).

18 Paulsen, J. et al. Long-range interactions between topologically associating domains shape the four-dimensional genome during differentiation. *Nat. Genet.* **51**, 835–843 (2019).

19 Siersbæk, R. et al. Dynamic rewiring of promoter-anchored chromatin loops during adipocyte differentiation. *Mol. Cell* **66**, 420–435 (2017).

20 Tohyama, S. & Kobayashi, E. Age-Appropriateness of Porcine Models Used for Cell Transplantation. *Cell Transplant.* **28**, 224–228 (2019).

21 Rosen, E. D. & Spiegelman, B. M. What we talk about when we talk about fat. *Cell* **156**, 20–44 (2014).

22 Kleinert, M. et al. Animal models of obesity and diabetes mellitus. *Nat. Rev. Endocrinol.* **14**, 140–162 (2018).

23 Yang, C.-H. et al. Independent phenotypic plasticity axes define distinct obesity subtypes. *Nat. Metab.* **4**, 1150–1165 (2022).

24 Loos, R. J. & Yeo, G. S. The genetics of obesity: from discovery to biology. *Nat. Rev. Genet.* **23**, 120–133 (2022)

REVIEWERS' COMMENTS

Reviewer #1 (Remarks to the Author):

I would like to thank the authors for their detailed and thorough response. After carefully reviewing their comments and the revised text, I believe that they have correctly addressed the issues I raised.

A few comments below:

- Point 1-2 about a sequencing depth being a possible confounding factor: no visible effect of the library size on TAD size, IS, loop size distribution (new Sup Fig 2) => Issue now addressed, no more concern.
- Point 1-3 about cellular composition effect: the new deconvolution-derived analysis that focuses on a subset with similar cell type compositions provides a useful and necessary contribution to the study.
- Point 1-4 to 1-11 on the treatment vs. AT effect: the new Fig 1j-l and Sup Fig 8 successfully make the group effect more visible. Statements about this effect have been revised in the text as required.
- Point 1-12: Methods have been completed as needed.

I have nothing to add on the other points, all correctly addressed in my opinion. Thanks to the other reviewers for their contributions.

Reviewer #2 (Remarks to the Author):

The authors have addressed the concerns I have raised.

Reviewer #3 (Remarks to the Author):

The authors have addressed all the points that I am concerned, so I suggest to accept it.

Reviewer #1

Comment 1-1:

I would like to thank the authors for their detailed and thorough response. After carefully reviewing their comments and the revised text, I believe that they have correctly addressed the issues I raised.

A few comments below:

- Point 1-2 about a sequencing depth being a possible confounding factor: no visible effect of the library size on TAD size, IS, loop size distribution (new Sup Fig 2) => Issue now addressed, no more concern.

- Point 1-3 about cellular composition effect: the new deconvolution-derived analysis that focuses on a subset with similar cell type compositions provides a useful and necessary contribution to the study.

- Point 1-4 to 1-11 on the treatment vs. AT effect: the new Fig 1j-l and Sup Fig 8 successfully make the group effect more visible. Statements about this effect have been revised in the text as required.

- Point 1-12: Methods have been completed as needed.

I have nothing to add on the other points, all correctly addressed in my opinion.

Thanks to the other reviewers for their contributions.

Response 1-1:

We sincerely appreciate your comments and advice for improving the manuscript.

Reviewer #2**Comment 2-1:**

The authors have addressed the concerns I have raised.

Response 2-1:

We sincerely appreciate your comments and advice for improving the manuscript.

Reviewer #3:

Comment 3-1:

The authors have addressed all the points that I am concerned, so I suggest to accept it.

Response 3-1:

We sincerely appreciate your comments in improving the manuscript.